# Microbial genetic and transcriptional contributions to oxalate degradation by the gut microbiota in health and disease

Menghan Liu[1,2†], Joseph C Devlin[1,2], Jiyuan Hu[1], Angelina Volkova[1,2], Thomas W Battaglia[1], Melody Ho[1], John R Asplin[3], Allyson Byrd[4], P'ng Loke[1], Huilin Li[1], Kelly V Ruggles[1], Aristotelis Tsirigos[1], Martin J Blaser[5*], Lama Nazzal[1*]

[1]NYU Langone Health, New York, United States; [2]Vilcek Institute of Graduate Biomedical Sciences, New York, United States; [3]Litholink Corporation, Laboratory Corporation of America Holdings, Chicago, United States; [4]Department of Cancer Immunology, Genentech Inc, South San Francisco, United States; [5]Center for Advanced Biotechnology and Medicine, Rutgers University, New York, United States

**Abstract** Over-accumulation of oxalate in humans may lead to nephrolithiasis and nephrocalcinosis. Humans lack endogenous oxalate degradation pathways (ODP), but intestinal microbes can degrade oxalate using multiple ODPs and protect against its absorption. The exact oxalate-degrading taxa in the human microbiota and their ODP have not been described. We leverage multi-omics data (>3000 samples from >1000 subjects) to show that the human microbiota primarily uses the type II ODP, rather than type I. Furthermore, among the diverse ODP-encoding microbes, an oxalate autotroph, *Oxalobacter formigenes*, dominates this function transcriptionally. Patients with inflammatory bowel disease (IBD) frequently suffer from disrupted oxalate homeostasis and calcium oxalate nephrolithiasis. We show that the enteric oxalate level is elevated in IBD patients, with highest levels in Crohn's disease (CD) patients with both ileal and colonic involvement consistent with known nephrolithiasis risk. We show that the microbiota ODP expression is reduced in IBD patients, which may contribute to the disrupted oxalate homeostasis. The specific changes in ODP expression by several important taxa suggest that they play distinct roles in IBD-induced nephrolithiasis risk. Lastly, we colonize mice that are maintained in the gnotobiotic facility with *O. formigenes*, using either a laboratory isolate or an isolate we cultured from human stools, and observed a significant reduction in host fecal and urine oxalate levels, supporting our in silico prediction of the importance of the microbiome, particularly *O. formigenes* in host oxalate homeostasis.

**\*For correspondence:**
dye@mpi-cbg.de (ML);
martin.blaser@cabm.rutgers.edu
(MJB);
Lama.Nazzal@nyulangone.org
(LN)

**Present address:** †Department of Biological Sciences, Columbia University, New York, United States

## Introduction

Over-accumulation of oxalate in humans leads to toxicity (*Asplin et al., 1998*; *Beck et al., 2013*). The most common oxalate toxicity is calcium oxalate nephrolithiasis, which accounts for more than 70% of overall nephrolithiasis, affecting 9% of the US population with a 20% 5-year recurrence rate (*Lieske et al., 2014*; *Rule et al., 2014*; *Saran et al., 2018*; *Scales et al., 2012*; *Stamatelou et al., 2003*). Oxalate toxicity can also induce chronic kidney disease, an illness affecting more than 30 million Americans, via multiple mechanisms including the activation of the NALP3 inflammasome pathway (*Knauf et al., 2013*; *Mulay et al., 2017*; *Mulay et al., 2013*; *Saran et al., 2018*; *Waikar et al., 2019*), RIPK3-MLKL-mediated necroptosis (*Mulay et al., 2016a*), and oxidative stress-induced cell injury (*Khan et al., 2006*). In extreme cases, life-threatening systemic oxalosis occurs.

Humans lack endogenous oxalate-degrading enzymes. By contrast, the mammalian intestinal microbes can degrade oxalate, partially protecting their hosts against toxicity (*Allison and Cook, 1981*; *Allison et al., 1986*; *Allison et al., 1977*; *Barber and Gallimore, 1940*; *Miller et al., 2014*; *Azcarate-Peril et al., 2006*) *Oxalobacter formigenes*, Lactobacillus sp., Bifidobacterium sp., Enterobacteriaceae, and others can degrade oxalate in vitro (*Mogna et al., 2014*), and colonization with these taxa in rodent hyperoxaluria models showed reduction in urinary oxalate indicating oxalate degradation in vivo (*Hatch et al., 2006*; *Klimesova et al., 2015*; *Kwak et al., 2006*). Of these oxalate degraders, only *O. formigenes* is a specialist that uses oxalate as its sole energy source (*Cornick and Allison, 1996*). *O. formigenes* also is unique because it induces host oxalate secretion into the colonic lumen (*Arvans et al., 2017*).

The oxalate-degrading microbes in the human microbiota in vivo have not been characterized. Previous studies on oxalate-degrading microbes have been chiefly done in vitro or in animal models, and the relevance of those microbes to human health remains undefined (*Fontenot et al., 2013*; *Klimesova et al., 2015*; *Kullin et al., 2014*; *Turroni et al., 2010*; *Turroni et al., 2007*). This gap limits our understanding of the role of microbiota in diseases induced by oxalate toxicity. Here, we leveraged multi-omics data of the healthy human microbiome to characterize the oxalate-degrading microbes in vivo.

Inflammatory bowel disease (IBD) patients are at increased risk for oxalate toxicity, due to a condition called enteric hyperoxaluria (EH). In EH, enhanced bioavailability and hyperabsorption of intestinal oxalate result in oxalate nephrolithiasis (*Corica and Romano, 2016*; *Liu and Nazzal, 2019*). In the USA, >50,000 IBD patients suffer from EH and recurrent calcium oxalate kidney stones (*Corica and Romano, 2016*; *McConnell et al., 2002*). EH in IBD patients may reflect lipid malabsorption and increased gut permeability. However, an alternate hypothesis is that microbiota-based oxalate degradation is impaired in dysbiotic IBD patients, leading to increased oxalate absorption (*Allison et al., 1986*). We interrogated the microbiota of IBD patients to understand shifts in microbiota-based oxalate degradation functions and their metabolic consequences.

## Results

### Type I and type II microbial oxalate degradation pathways

To determine the oxalate degradation pathways (ODPs) used by human gut bacteria, we curated all experimentally validated microbial ODP from literature review and database searches (see Materials and methods) (*Allison et al., 1985*; *Anand et al., 2002*; *Blackmore and Quayle, 1970*; *Daniel et al., 2004*; *Dumas et al., 1993*; *Foster et al., 2012*; *Pierce et al., 2010*). We classified those ODPs into two types based on their enzymatic mechanisms and co-factor requirements. Type I ODPs cleave the oxalate carbon-carbon (C-C) bond in a single step (*Figure 1A*). The two recognized type I enzymes, oxalate oxidase and oxalate decarboxylase, are indistinguishable at the amino acid level; therefore, we refer to them jointly as oxalate oxidase/decarboxylase (OXDD) (*Figure 1A*). Type II ODPs consists of two enzymatic reactions requiring coenzyme A as co-factor (*Figure 1A*). First, a coenzyme A molecule is added to oxalate to form oxalyl-CoA via enzymes including formyl-CoA transferase (FRC) (*Figure 1A*). In the second step, oxalyl-CoA decarboxylase (OXC) metabolizes oxalyl-CoA into $CO_2$ and formyl-CoA (*Figure 1A*).

Knowing the relevant oxalate degradation enzymes (ODEs), we then acquired all available protein homologs of the three ODE OXDD (n = 2836), FRC (n = 1947), and OXC (n = 1284), which enable homology search from UniProt Interpro (*Mitchell et al., 2019*; *Mulder et al., 2005*). By tracing the taxonomic origin of the genes encoding those homologs, we found that OXDD-coding taxa can be fungal or bacterial, whereas FRC- and OXC-coding taxa are strictly bacterial (*Figure 1B*). The frequent co-occurrence of FRC and OXC in individual genomes indicates encoding complete type II ODP (*Figure 1B*). As expected, OXDD, FRC, and OXC each are conserved within the same microbial class, but exhibit substantial divergence across classes (*Figure 1—figure supplement 1*). These data provide both a comprehensive inventory of ODPs and a reference set of ODP-encoding microbes, which enable analyses to elucidate those relevant to humans.

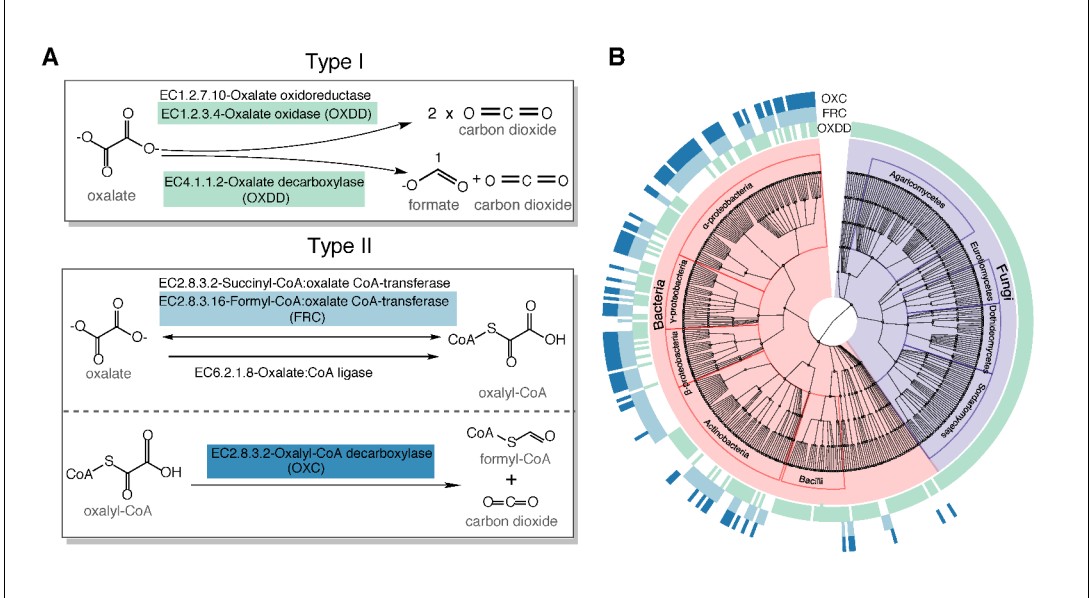

**Figure 1.** Type I and type II microbial oxalate-degrading pathway (ODP). (**A**) Schema of type I and type II ODP. Enzymes are annotated with corresponding KEGG IDs. OXDD, FRC, and OXC are the focus of the present study. (**B**). Cladogram of microbial genera that encode oxalate-degrading enzymes OXDD, FRC, and OXC. The three rings surrounding the cladogram indicate OXDD-, FRC-, or OXC-encoding genera, respectively. The online version of this article includes the following figure supplement(s) for figure 1:

**Figure supplement 1.** Inter-class and intra-class ODE protein identity associated with each microbial class.

## ODPs utilized by the gut microbiota of healthy humans

Next, we asked whether those ODPs are encoded or expressed by microbes within the intestinal tract of healthy humans. To do so, we computationally examined the presence of the ODE within the gut metagenome and metatranscriptome from publicly available samples of healthy humans. From five studies, we analyzed a total of 2359 metagenome and 1053 transcriptome samples from 660 and 165 healthy individuals, respectively (*Figure 2—figure supplement 1*, *Supplementary file 1a*). After quality-filtering (see Materials and methods), the sequencing reads were aligned to the unique OXDD, FRC, and OXC homologous proteins we had identified, using DIAMOND Blastx (*Buchfink et al., 2015*). Alignment pairs with >90% identity were retained for downstream analyses. The alignment cutoff was based on the protein identity of the inter- and intra-species ODEs and determined to be robust for distinguishing ODEs originating from differing microbial species (*Figure 2—figure supplement 2*).

We found that the majority of the healthy gut microbiomes include ODEs with at least one ODE detected in the metagenome of 607 (92%) of 660 subjects and the metatranscriptome of 132 (80%) of 165 subjects. In the metagenomes, the type II *frc* and *oxc* genes were more common (*Figure 2A*) and more abundant (*Figure 2B*) than the type I *oxdd* gene. Similarly, in metatranscriptomes, expressions of type II genes were more common and abundant (*Figure 2C,D*). Expression of *oxdd* was only detected in 10 (6%) of the 165 subjects, and the median RPKM was 2-$\log_{10}$ lower than those of *frc* or *oxc* (*Figure 2C,D*). Furthermore, *frc* and *oxc* – the coding genes of the two enzymes that catalyze the separate steps in type II PDP – were frequently co-expressed within the same microbiota, indicating expression of the complete type II ODPs (*Figure 2—figure supplement 3*).

These data indicate that microbes utilizing type II rather than type I ODPs predominate in the human intestine. Such finding is consistent in all studies, despite the differences in source populations and sample preparation methods (*Supplementary file 1a*). For the remaining analyses, we focused on the type II ODPs.

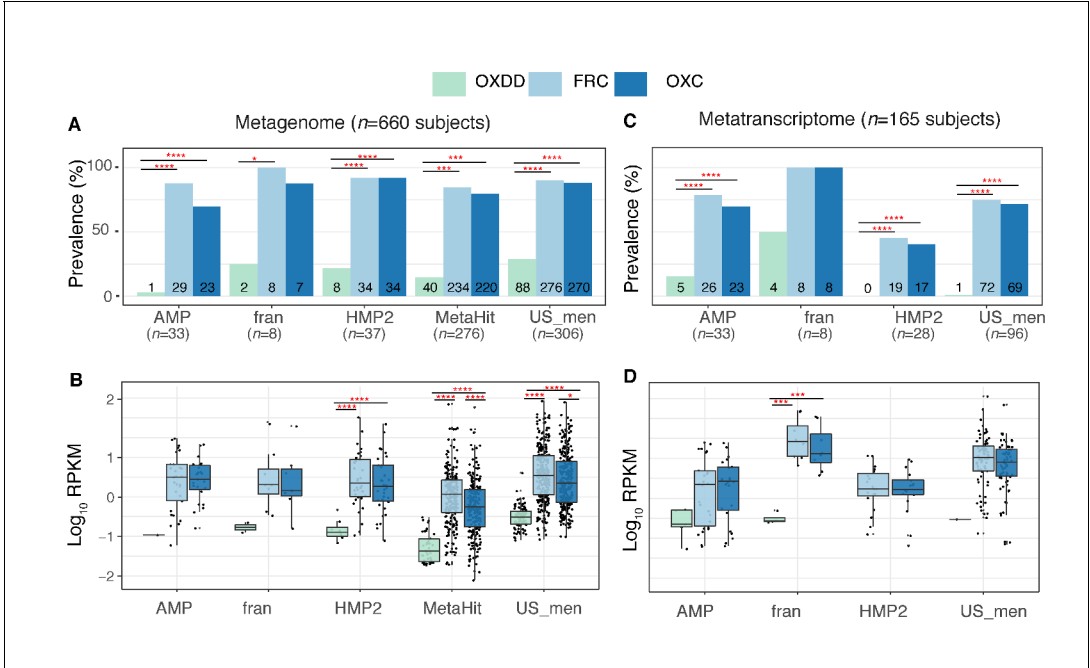

**Figure 2.** Detection of type I and II ODE within the fecal metagenome and metatranscriptome of 660 and 165 healthy human subjects. Prevalence (A) and abundance (B) of ODE in the fecal metagenome of five studies surveyed. Numbers written on the bottom bars indicate the numbers of subjects in whom the corresponding ODE is detected, and only those subjects were considered in (B). Prevalence (C) and abundance (D) of OXDD, FRC, and OXC in the fecal metatranscriptome of four studies surveyed. *p<0.05, **p<0.01, ***p<0.001, ****p<0.0001, by proportion tests for (A) and (C), by multiple-adjusted Mann–Whitney tests for (B) and (C).

The online version of this article includes the following source data and figure supplement(s) for figure 2:

**Source data 1.** Detection of OXDD, FRC, and OXC in the metagenome and metatranscriptome of healthy individuals.

**Figure supplement 1.** Beta-diversity of metabolic profiles associated with the metagenomic and metatranscriptomic samples from healthy human subjects, ordinated on a Tsne (t-distributed stochastic neighbor embedding) plot.

**Figure supplement 2.** The protein identity between inter-species and intra-species ODEs, for each microbial genus.

**Figure supplement 3.** Co-detection of OXDD, FRC, and OXC in the metatranscriptomes of subjects across different studies.

## Microbial species that transcribe the type II ODPs in vivo

Although multiple human commensal microbes are known to encode *frc* and *oxc*, whether they transcribe those genes in vivo has not been studied (*Abratt and Reid, 2010*; *Cho et al., 2015*; *Fontenot et al., 2013*; *Mogna et al., 2014*). We next characterized the microbial species transcribing these type II ODP genes in the microbiota of healthy humans. In the metagenomes of 660 individuals, *oxc* gene of multiple species, including *Escherichia coli*, *O. formigenes*, and several Muribaculaceae, Bifidobacterium, and Lactobacillus sp., was detected (*Figure 3A*, left); *E. coli oxc* was the most common (56% of subjects), followed by *O. formigenes oxc* (39% of subjects).

In the metatranscriptomes of 167 individuals, *oxc* gene expression did not directly correlate with corresponding gene abundance or prevalence (*Figure 3A*, right). *O. formigenes oxc* expression was both most abundant and most prevalent (in 61% of subjects) in the metatranscriptomes (*Figure 3A*, right). Despite the detection of *E. coli oxc* in 56% of subjects, its transcript was present in only 12% of the subjects (*Figure 3A*). For Bifidobacterium and Lactobacillus species, for which oxalate degradation activity was reported in vitro and in animal models (*Federici et al., 2004*; *Klimesova et al., 2015*; *Turroni et al., 2010*), *oxc* expression was minimal (<5%) or not prevalent (*Figure 3A* right). The dichotomy between metagenomic and metatranscriptomic *oxc* was consistent across different studies (*Figure 3—figure supplement 1*, *Figure 3—figure supplement 2*), and also present for the other type II ODP gene *frc* (*Figure 3—figure supplement 3*).

To more rigorously examine ODP expression by individual taxa, we further co-analyzed the presence of ODP genes and transcripts matched by subject (*Figure 3B*). *O. formigenes frc* and *oxc* were

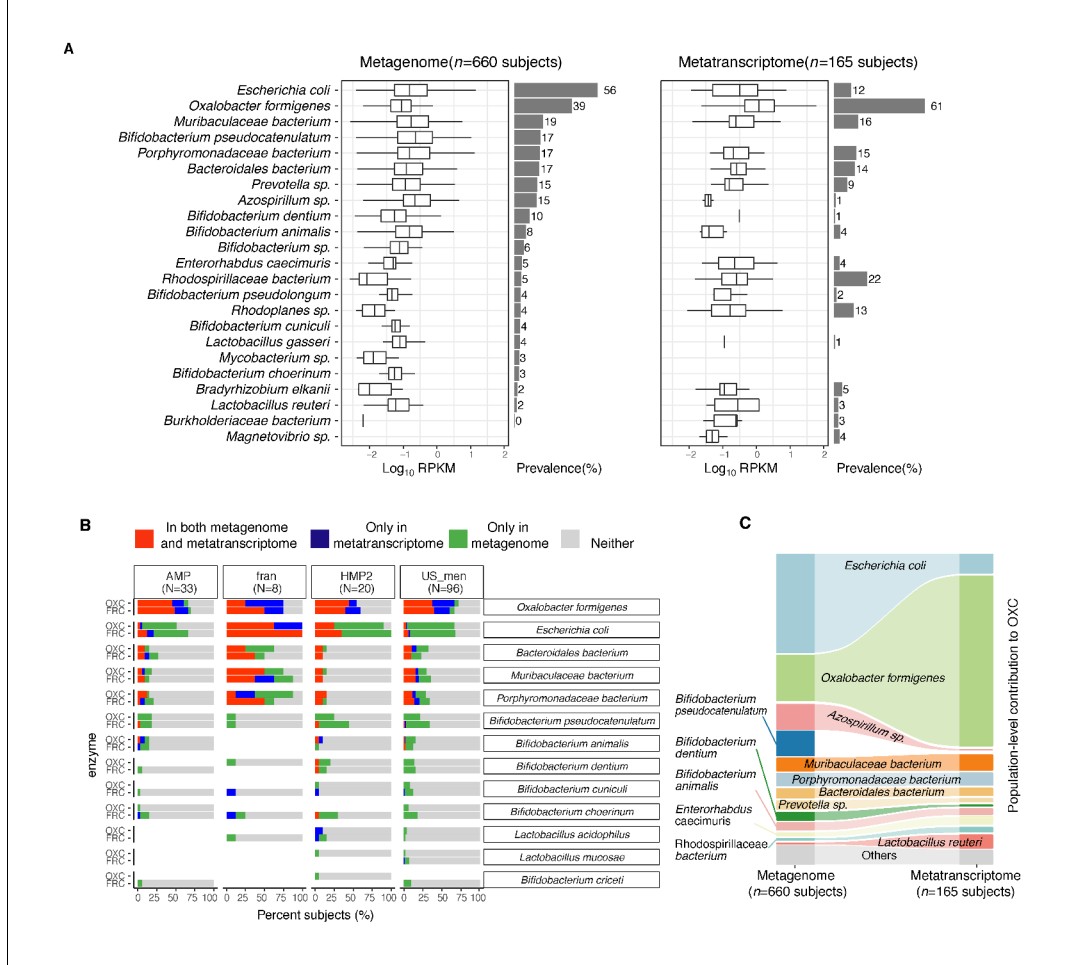

**Figure 3.** Expression of type II ODP of microbial species within the intestinal microbiota of healthy humans. (A) Abundance and prevalence of OXC of microbial species in the metagenome (left) or metatranscriptome (right) of 660 and 165 subjects. Box plots indicate the abundance of microbial OXC (log$_{10}$ RPKM) among subjects in whom OXC is detected, and are generated with *ggplot2* with outliers excluded. Bar plots indicate the prevalence of microbial oxc, with percentage annotated. Microbial species are ordered by the corresponding metagenomic OXC prevalence. A parallel analysis for FRC is shown in *Figure 3—figure supplement 5*. (B) Detection of OXC and FRC of microbial species in the subject-matched metagenome and metatranscriptome, by study. For each microbial ODE, the subjects are divided into four groups (shown in different colors) based on the co-detection of ODE in the matched metagenome and metatranscriptome, with percent (%) of which reflected. The fran Study, from which *E. coli* ODP was detected in all subjects, used a sample extraction method known to induce *E. coli*, as noted in their publication (*Franzosa et al., 2014*). (C) Population-level contribution of individual species to metagenomic (left) or metatranscriptomic (right) OXC. The population-level contribution of each species was calculated at a relative scale (see Materials and methods) and plotted. Raw values can be found in *Supplementary file 1a*. The 10 species that have the highest metagenomic or metatranscriptomic contribution are shown. A parallel analysis for FRC is shown in *Figure 3—figure supplement 3*.

The online version of this article includes the following source data and figure supplement(s) for figure 3:

**Source data 1.** Species contribution to FRC, and OXC in the metagenome and metatranscriptome of healthy individuals.

**Figure supplement 1.** Detection of OXC of microbial species in the microbiome of healthy human subjects from US_men (A), HMP2 (B), AMP (C), or fran (D) study.

**Figure supplement 2.** Tsne plot of 594 metagenomic and 131 metatranscriptomic samples, based on the abundances OXC and FRC.

**Figure supplement 3.** Detection of FRC of microbial species in the metagenome (left) or metatranscriptome (right) of healthy human subjects.

**Figure supplement 4.** Population-level contribution of individual species to metagenomic (left) or metatranscriptomic (right) FRC.

**Figure supplement 5.** Detection of ODE using ShortBRED.

**Figure supplement 6.** Detection of *O. formigenes* OXC using merged long marker peptides identified by ShortBRED.

transcribed in nearly all subjects in whom the genes were detected, as well as in others in whom the gene was not detected, indicating that their expression is common in vivo (*Figure 3B*). The under-detection of *O. formigenes* ODP genes in the metagenomes may reflect the highly variable abundance of the organism, often below the lower detection limit using gene-based methods

(*Barnett et al., 2016*; *Guo et al., 2017*; *Kelly et al., 2011*; *PeBenito et al., 2019*). In contrast, *E. coli frc* and *oxc* were expressed in only a few subjects even when the corresponding genes were detected metagenomically (*Figure 3B*). These data indicate that in vivo *E. coli* rarely transcribe ODP, a pathway used for defense against oxalate-induced stress (*Fontenot et al., 2013*). In total, these findings demonstrate that ODP transcription varies widely in individual hosts, and by species.

## The contributions of individual species to the global microbiota ODP

We then assessed the impact of individual species on global ODP by quantifying their population-level contributions (see Materials and methods). The contribution of *O. formigenes* to ODP increased from 17% to 63% from the metagenomic to the metatranscriptomic level, greater than the transcriptomic contributions of all other species combined (*Figure 3C*, *Supplementary file 1b*). Conversely, the *E. coli* contribution to ODP was markedly reduced from the metagenomic (36%) to the metatranscriptomic (7%) level (*Figure 3C*). Other species had low but varied contributions (*Figure 3C*). A parallel pattern was observed for *frc* (*Figure 3—figure supplement 4*, *Supplementary file 1c*). With the low activity of non-*O. formigenes* species, network analysis did not yield significant species-species interactions related to *oxc* transcription (data not shown). In summary, we found that the type II ODP genes, *frc* and *oxc*, are encoded by multiple gut microbes, but *O. formigenes* dominated this pathway at the transcriptional level. These data provide a baseline for examining disease-associated changes.

## Validation of ODP detection using ShortBRED

ShortBRED (*Kaminski et al., 2015*) is a tool for microbiome functional profiling, which clusters protein homologs into clusters, and identifies marker peptide for each cluster, thus potentially achieving high specificity. We reanalyzed all samples using ShortBRED to validate our bioinformatics findings (see Materials and methods). Based on ShortBRED, FRC and OXC were significantly more abundant and more prevalent than OXDD; their abundances were significantly correlated with our previous results (*Figure 3—figure supplement 5A*). Consistently, *O. formigenes* was the species with the highest transcriptional activity for FRC (*Figure 3—figure supplement 5B*). Specifically, each of the three *O. formigenes* FRC homologs (C3 × 9Y2, C3 × 762, and C3 × 2D4) are distinct from other homologs and from each other; thus, each formed a singleton family (*Supplementary file 2*) with unique peptide markers (*Supplementary file 3*). The three *O. formigenes* FRCs are the most commonly transcribed among FRCs encoded by any taxon. They are present in the metatranscriptome of 50, 52, and 41% of the tested subjects, respectively (*Figure 3—figure supplement 5B*, right), and also are among the most abundant (ranks 1, 2, and 5) among all FRC proteins (*Figure 3—figure supplement 5B*, right).

The taxonomic contributions based on BURRITO (*McNally et al., 2018*) (see Materials and methods) (*Supplementary file 1d*) are highly concordant with findings using our prior method (*Supplementary file 1b*, metagenome, Healthy). *E. coli* and *O. formigenes* are consistently the two largest contributors, and several Bifidobacterium spp and Lactobacillus spp contributed to a lesser extent.

ShortBRED did not detect *O. formigenes* OXC in any sample. We reasoned that this under-detection is due to poorly-selected marker peptides. Indeed, for the protein cluster of *O. formigenes* OXC (consists of two *O. formigenes* OXC homologs: C3XBB9 and C3 × 545, with C3 × 545 as the centroid), the makers selected are as follows: (1)short (median length 16.2 amino acids-in comparison, the marker for *O. formigenes* FRC is 300 amino acids) and (2) improperly broken up from long continuous regions (*Figure 3—figure supplement 6A*, Marker #1–20). Therefore, we manually generated new markers (Markers #21–24) by combining markers that were one amino acid apart. Using these longer markers, ShortBRED successfully detected OXC using three of the four markers (except for the shortest one) with the expected patterns (*Figure 3—figure supplement 6B,C*).

In summary, all of our major conclusions examined were confirmed with ShortBRED. The diamond-mapping method utilizes reference full length and relies on best alignment scores to assign reads with higher sensitivity, while ShortBRED uses unique regions of reference with higher specificity. Because of their different advantages, the two methods are complementary and both were used for downstream analyses.

## Increased enteric oxalate levels and reduced microbial ODP expression in IBD patients

IBD patients, particularly patients suffering from CD, frequently have EH, with oxalate hyperabsorption and calcium oxalate nephrolithiasis (*Corica and Romano, 2016*; *Liu and Nazzal, 2019*; *McConnell et al., 2002*). Ulcerative colitis (UC), regardless of severity and location, is associated with stone formation (*Cury et al., 2013*), but ileocolonic CD is associated with greater nephrolithiasis risk than either ileal or colonic involvement alone (*Cury et al., 2013*). We hypothesized that oxalate degradation by the intestinal microbiome may be impaired in IBD patients, leading to more luminal oxalate in the host available for passive absorption.

We tested this hypothesis, using the multi-omics data of IBD patients and healthy subjects from the iHMP-IBD study (*Franzosa et al., 2019*; *Lloyd-Price et al., 2019*). The patients were stratified by illness: UC (N = 30 subjects) and CD (N = 54), and the CD group was further divided into CD-L3 (N = 25), with ileocolonic phenotype at baseline and CD-nonL3 (N = 29) without, as defined by the L3 IBD Montreal classification (*Satsangi et al., 2006*). Consistent with the clinical nephrolithiasis risk, fecal oxalate relative abundances were elevated in both the UC (p=0.005) and CD (p=0.06) patients compared to healthy controls (*Figure 4A*). All of the CD risk was in the CD-L3 patients (p<0.001), and not in the non-CD-L3 patients, indicating that IBD location, particularly ileocolonic involvement, is key for EH risk (*Figure 4A*, *Figure 4—figure supplement 1*). The fecal oxalate levels were not clearly different in relation to disease activity (*Damms and Bischoff, 2008*; *Jowett et al., 2001*; *Manz et al., 2012*; *Pathirana et al., 2018*; *Figure 4—figure supplement 2*). We observed increased fecal oxalate in association with higher inflammation levels (fecal calprotectin > 50 µg/g) (*Figure 4—figure supplement 2C*); since our number of study subjects was small, this relationship needs to be examined with larger patient cohorts in future studies.

The global transcripts of *frc* and *oxc* were reduced in the IBD patients compared to the controls in all four studies analyzed (*Figure 4B, Figure 4—figure supplement 3*). Expression of *oxc* was detected in 57% UC, 54% CD and 48% CD-L3 subjects, lower than the 71% in healthy individuals (*Figure 4B*). ODP expression was least impacted in the CD-nonL3 group (*Figure 4B, Figure 4—figure supplement 3*), in which the fecal oxalate increase was not observed (*Figure 4A*). The total transcripts of *frc* and *oxc* were also significantly lower in IBD patients (*Figure 4C*, *Figure 4—figure supplement 3B*) (p<0.001 for *frc* for all groups compared to healthy, p<0.01 for *oxc* in UC patients compared to healthy). We also observed that *frc* expression was significantly inversely correlated with oxalate relative abundance in the UC group (*Figure 4—figure supplement 4A*). The same inverse trend was marginally observed for *oxc* but was not statistically significant (*Figure 4—figure supplement 4B*); thus, this question will need to be examined in larger future cohorts. In contrast to reduced *frc* and *oxc* transcripts in IBD patients, the *frc* and *oxc* genes were significantly more abundant (*Figure 4—figure supplement 5*), indicating that the *frc*- and *oxc*-encoding taxa (e.g. Enterobascteriaceae) are enriched in the IBD gut but do not actively express ODP.

Collectively, these data showing the reduction of ODP-associated transcripts in the IBD patient microbiota suggest their role in the elevated intestinal oxalate levels, and possibly increased susceptibility to nephrolithiasis.

## Loss of *O. formigenes* and its ODP-associated transcripts in IBD patients

We next sought to identify the microbial species accounting for the reduced ODP transcripts in the IBD patients. *E. coli* and *O. formigenes* with the largest ODP contributions at the genomic and transcriptional level, respectively, were notable. Using gene and transcript jointly as markers, ODP expression by *O. formigenes* was detected in ~25% of UC and CD patients (*Figure 5A*), significantly less than in healthy persons (~70%), either when studies were combined (*Figure 5A*) or separate (*Figure 3B*). In contrast, *E. coli* ODP was detected in nearly all IBD subjects and was transcribed more frequently compared to healthy subjects (*Figure 5A*). Consistent with the low overall prevalence, transcripts for *O. formigenes* ODP expression were less abundant (lower RPKM values) in UC, CD, and in CD-L3 patients compared with controls (p<0.01 for all groups) (*Figure 5B*). In the IBD subjects, the observed *O. formigenes* ODP genes were always actively expressed (*Figure 5A*). Significantly elevated fecal calprotectin levels were observed when *O. formigenes* was absent in healthy individuals and in CD-L3 patients (*Figure 4—figure supplement 2D*).

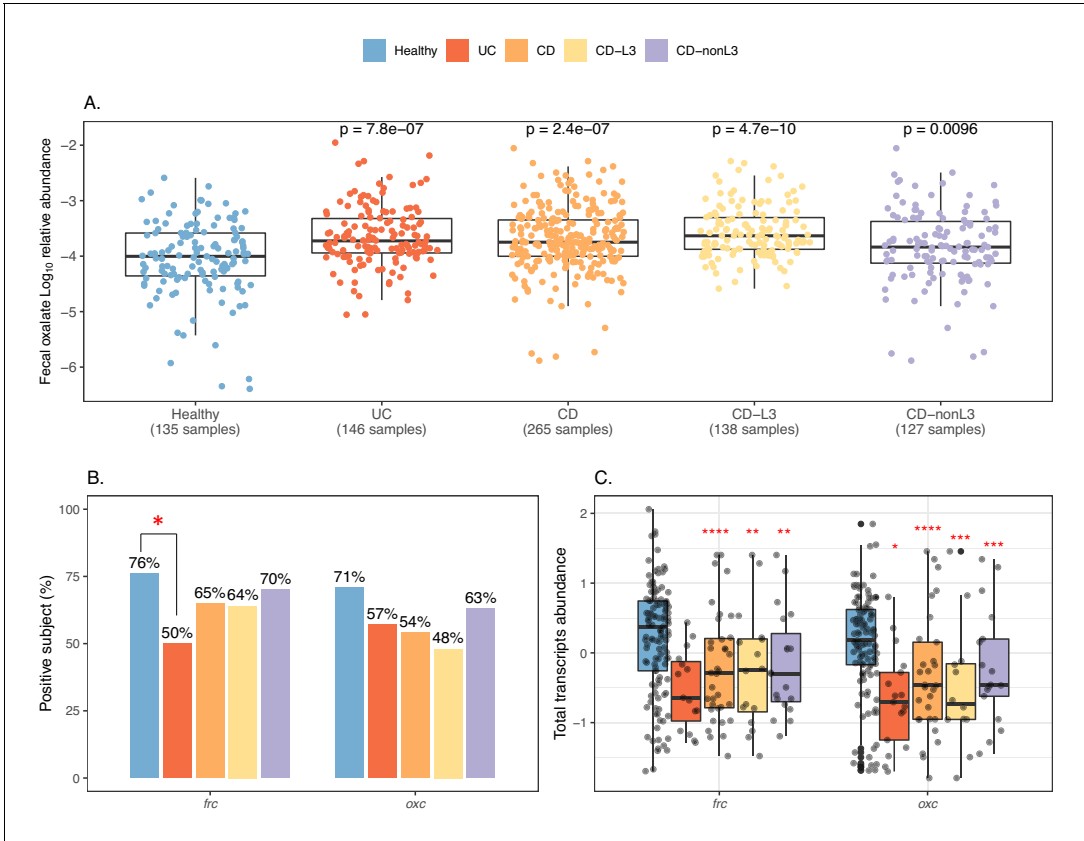

**Figure 4.** Elevated fecal oxalate and reduced expression of microbiome ODP in IBD patients. (**A**). Stool oxalate relative abundance (log10) in healthy, UC, CD, CD-L3, or CD-nonL3 subjects from HMP-IBD study. Fecal oxalate relative abundance was determined from untargeted metabolomics data from the iHMP-IBD; measurements related to oxalate were selected and normalized against total metabolites (percent abundance of all observed metabolites) for analysis. L3 refers to the ileocolonic phenotype, according to the Montreal Classification at baseline. Data derived from iHMP-IBD untargeted metabolomics measurements. Prevalence (**B**) and abundance (**C**) of OXDD, FRC, and OXC in metatranscriptomes of healthy, UC, CD, or CD-L3 subjects. The 165 healthy controls are combined from four studies (AMP, US_men, fran, HMP2). *: p<0.05, **: p<0.01, ***: p<0.001, ****: p<0.0001 by multiple-adjusted Mann-Whitney tests in (**A**) and (**C**), by proportion test in (**B**).

The online version of this article includes the following source data and figure supplement(s) for figure 4:

**Source data 1.** Fecal oxalate and ODE expression in healthy and IBD individuals.

**Figure supplement 1.** Fecal oxalate log10 relative abundance in CD patients, according to the Montreal clinical classification (*Satsangi et al., 2006*).

**Figure supplement 2.** Comparison of fecal oxalate log10 relative abundance based on disease activity by fecal calprotectin levels or SCCAI scores.

**Figure supplement 3.** Prevalence (**A**) and abundance (**B**) of FRC, and OXC in metatranscriptomes of healthy, UC, CD, or CD-L3 subjects detected by ShortBRED.

**Figure supplement 4.** Spearman correlations of fecal oxalate and total transcripts of frc (**A**) or oxc (**B**).

**Figure supplement 5.** Abundance of *frc* and *oxc* genes in the metagenome of IBD patients and healthy individuals.

In total, these data indicate that the absence of *O. formigenes* colonization or colonization below the level of detection is responsible for the reduction in global ODP transcripts. *E. coli*, and Lactobacillus and Bifidobacterium spp. use ODPs to defend against oxalate-induced acid stress; their upregulation in IBD appears secondary to the elevated oxalate levels present.

## Effect on *O. formigenes* colonization on the host urinary and fecal oxalate levels

To validate our bioinformatics prediction that *O. formigenes* is an important oxalate-degrading organism that can influence oxalate homeostasis in vivo, we examined whether the colonization of *O. formigenes* results in significant reduction in urinary oxalate in mice (*Figure 6A*). Mice from our gnotobiotic facility which had a total microbiota 2–3 log10 lower than conventional mice were fed with diet supplemented with 1% sodium oxalate and 0.5% calcium. The 1% dietary oxalate is lower

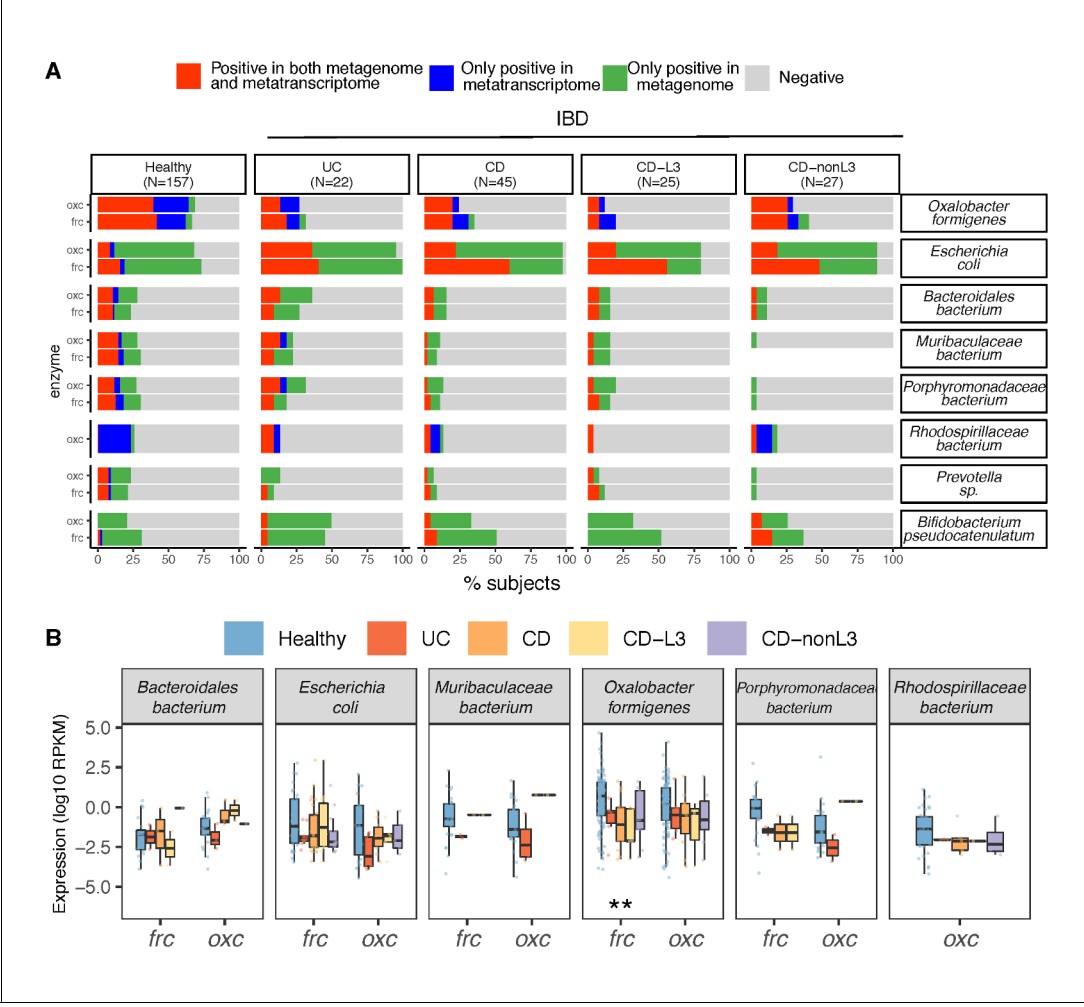

**Figure 5.** Differential ODP expression by human gut microbes in healthy and disease states. (**A**) Detection of microbial OXC and FRC in the subject-matched metagenome and metatranscriptome from healthy subjects, UC, CD, or CD-L3 patients. For each species shown, the subjects are divided into one of four categories based on the co-detection of ODE in the matched metagenome and metatranscriptome. (**B**) Expression of microbial FRC and OXC in the metatranscriptomes of healthy subjects, UC, CD, or CD-L3 patients. Boxplot reflects the subjects, in whose metatranscriptome the corresponding enzyme is detected. *p<0.01, **p<0.0001 by multiple-adjusted Mann–Whitney tests.

The online version of this article includes the following source data for figure 5:

**Source data 1.** Species contribution to FRC, and OXC in IBD individuals.xlsx.

than previous studies (*Hatch et al., 2006*; *Hatch et al., 2011*), in order to reduce chronic kidney damage described previously (*Mulay et al., 2016b*). This diet resulted in significant increase of oxalate (23.3% and 174.8% increase in fecal and urinary oxalate, respectively) compared to normal chow (data not shown). We then colonized these mice with either of two *O. formigenes* strains, including a widely studied lab strain (OXCC13) and a human *O. formigenes* isolate from the stool sample of a patient with primary hyperoxaluria type 1 (PH1) isolated as described (*Allison et al., 1985*). Continued colonization with each was established by qPCR (*Figure 6—figure supplement 1*). Using targeted oxalate assays, the *O. formigenes*-colonized mice had significantly lower urinary and fecal oxalate levels, compared to the non-colonized mice (*Figure 6B–C*); the values from the two measurements tended to correlate in individual mice (*Figure 6D*).

In our model, fecal *O. formigenes* averaged ~5×10$^6$ by qPCR (*Figure 6—figure supplement 1*). In contrast, *O. formigenes* was below the detection limit (<10$^2$) in the jejunum and ileum of most mice (*Figure 6—figure supplement 2*). These data suggest that *O. formigenes* primarily colonizes and degrades oxalate in the host colon, which is known to be an important site for oxalate absorption.

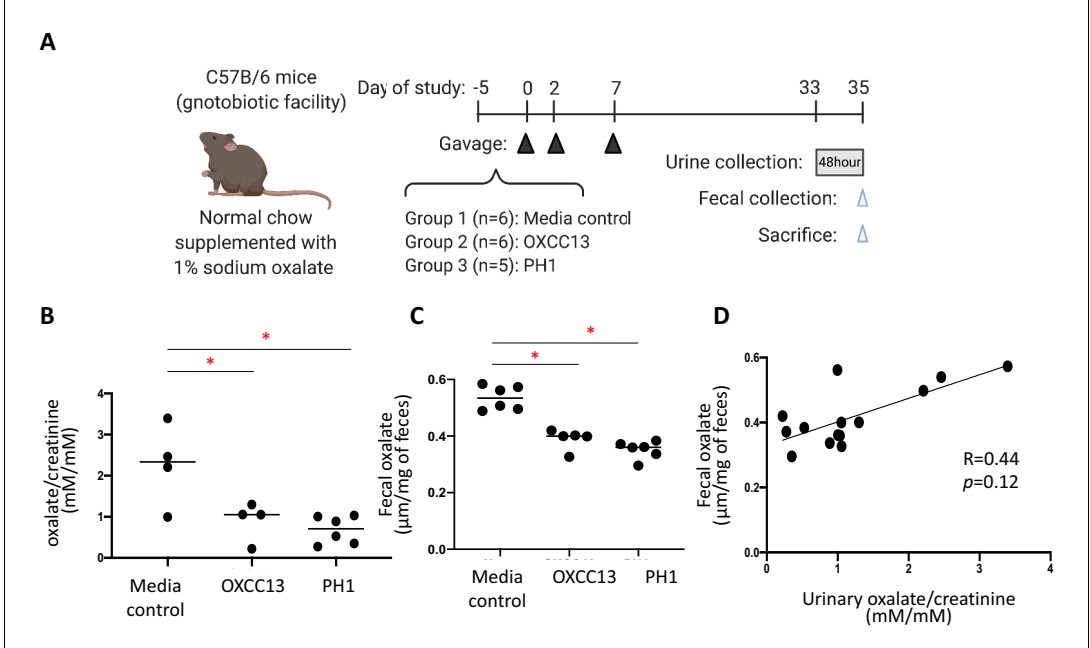

**Figure 6.** Effect of *O. formigenes* colonization on fecal and urinary oxalate. (**A**) Study design of the mouse experiment. C57Bl/6 mice from our gnotobiotic facility were assigned to three groups. At days 0, 2, and 7, mice were gavaged (blue arrowheads) with *O. formigenes* strain OXCC13 (n = 5), *O. formigenes* freshly isolated from a primary hyperoxaluria type 1 (PH1) subject (n = 6), or *O. formigenes* culture medium alone (Media). Mice were fed with normal chow supplemented with 1% sodium oxalate and 0.5% calcium from day −5 until sacrifice. Urine was obtained from a 48 hr collection (one to two mice per pool) prior to sacrifice, and feces were collected at sacrifice (blue arrowhead). (**B, C**) Urinary and fecal oxalate in three mouse groups. Urinary oxalate normalized by creatinine in the 48 hr urine samples and fecal oxalate levels per gram of stool samples in the three experimental groups. *$p<0.05$, by Tukey's multiple comparisons test. (**D**) Relationship between fecal and urinary oxalate. Correlation coefficient was computed using Spearman's r.

The online version of this article includes the following source data and figure supplement(s) for figure 6:

**Source data 1.** Fecal and urinary oxalate in relation to *O. formigenes* colonization in mice.

**Figure supplement 1.** Detection of *Oxalobacter formigenes* by qPCR in the mouse fecal samples before sacrifice.

**Figure supplement 2.** Detection of *O. formigenes* by qPCR in the intestinal contents (Jej: jejenum, Ile: ileum) and mouse fecal (Fec) samples at sacrifice.

Taken together, these findings suggest that *O. formigenes* is sufficient to reduce host urinary and fecal oxalate levels, and support our in silico prediction of the importance of *O. formigenes* to host oxalate homeostasis, with protection against oxalate-induced toxicity.

## Discussion

Oxalate degradation by the human microbiota has been known since the 1940s (*Allison and Cook, 1981*; *Allison et al., 1986*; *Allison et al., 1985*; *Allison et al., 1977*; *Barber and Gallimore, 1940*), but the taxa involved in vivo has not been systematically described. We present the first comprehensive study of human oxalate-degrading microbes and define their individual contributions. We successfully distinguished the taxa that are actively transcribing ODP from those that encode the pathway, but with low expression, by co-analyzing metagenome and metatranscriptome data.

Our finding that multiple human gut microbes encode ODPs is consistent with prior studies (*Abratt and Reid, 2010*; *Klimesova et al., 2015*; *Mogna et al., 2014*; *Stern et al., 2016*; *Ticinesi et al., 2018*). But surprisingly, at the transcriptional level, *O. formigenes* dominates the global ODP, which is consistent with *O. formigenes* being an oxalate autotroph, as well as FRC and OXC being the most abundant proteins during both exponential and lag stages (*Ellis et al., 2016*). These findings were consistent using our method and ShortBRED, which have their unique analytic strengths. The diamond-mapping method utilizes protein full-length information and is completely agnostic. By setting a stringent identity cutoff of 90%, we retain only the high-confidence alignment

pairs. The finding generated using this 90% cutoff is consistent with the observations generated through the second method ShortBRED, which uses ODE-specific marker peptides, which provide evidence that the cutoff is highly effective at preventing spurious alignments. By contrast, Short-BRED generated highly specific markers but is limited by several parameter choices (CD-Hit, centroid protein selection, length and identity of short identical regions and final marker). Because of their specific advantages, the two methods are complementary.

The contrasting genetic and transcriptional ODP differences we observed highlight the importance of analysis beyond the gene level for microbiome studies. There are two critical advantages of an approach that co-analyzes metagenome and metatranscriptome: (1) Detection of genes in the metagenome does not ensure that they are being actively utilized by the indicated taxon. Thus, analyzing metatranscriptomic data allows distinguishing the taxa that are actively contributing to a biological process by generating the relevant proteins. In contrast, for those taxa that are not, the analysis provides a new view of their metabolism, in that although they have the potential for using a certain pathway or metabolite, it is not active under the particular condition tested. (2) Having metatranscriptomic data or other functional readout is particularly useful for cross-comparisons of the microbiome of different cohorts (i.e., diseased versus healthy subjects, in humans or experimental animals). For example, in our study, ODP genes were increased in IBD patients, whereas transcripts were significantly reduced. Specifically, the increased ODP gene abundances were due to the over-representation of *E. coli* strains in IBD patients, which uncommonly transcribe ODP in vivo. In contrast, the decrease of oxalate-degrading gene expression is caused by the loss of *O. formigenes*, which is the dominant microbe that transcribes this pathway. Having observed the differential abundance for a gene does not necessarily indicate functional shifts, as gene abundance is driven by the most abundant taxa, which could be transcriptionally silent. As such, the transcriptional evidence is a better indicator for evaluation of microbiome functional differences. The integrative multi-omics analysis framework built for this study, (now deposited on Github), can be extended to a broad range of microbiome functions.

EH is frequent in IBD patients, particularly in CD, and in those who underwent Roux-en-Y gastric bypass; the latter population is optimal for study since pre- and post-treatment samples can be easily collected. In the present study, we examine IBD patients, a risk population for EH and nephrolithiasis, and show that the impaired metabolic activity of the microbiota is correlated with enteric oxalate levels, thus nephrolithiasis risk. That fecal oxalate relative abundances were elevated in the iHMP-IBD UC patients, and in the CD patients with ileocolonic involvement, is consistent with their high nephrolithiasis risk (*Cury et al., 2013*; *Hylander et al., 1979*; *Hylander et al., 1978*). In this population at risk for EH, the ODP gene abundances are increased, but the extent of gene expression is decreased. This contrast between gene abundance and expression is due to differential ODP transcription in particular microbes.

IBD patients are known to have low levels of *O. formigenes* colonization (*Kumar et al., 2004*). The oxalate-degrading specialist *O. formigenes* is a gram-negative anaerobe susceptible to multiple antibiotics including macrolides, tetracyclines, metronidazole, and clindamycin (*Kharlamb et al., 2011*; *Mittal et al., 2005*). Thus, the reduced colonization of *O. formigenes* in IBD patients might reflect the frequent antibiotic treatments they receive. Human data showed that a single course of antibiotics to eradicate *Helicobacter pylori* results in the long-term suppression of *O. formigenes* colonization (*Kharlamb et al., 2011*). This supports the notion that antibiotics exacerbate microbiome dysbiosis and may lead to secondary conditions such as EH. These findings raise a further potential cost of antibiotic treatment in IBD patients, which should be weighed against potential benefits, especially when infection is not clearly demonstrated. Other changes in the intestinal milieu of IBD patients affecting pH, oxygen, and bile acid levels have been shown to impact *O. formigenes* in vitro (*Duncan et al., 2002*; *Allison et al., 1985*).

Our findings that *O. formigenes* is the main contributor to oxalate degradation in the healthy state, but is diminished in the IBD population, provides a strong rationale for *O. formigenes*-based restoration therapy. Restoring *O. formigenes* has been tested in primary hyperoxaluria (PH) patients, but yielded mixed results (*Hoppe et al., 2011*; *Hoppe et al., 2017*). However, PH patients might not be the best subjects for restoration because their oxalate toxicity is caused by hepatic overproduction of oxalate, and microbial degradation is restricted to the oxalate secreted into the gut lumen. In EH, the oxalate overload is in the intestinal lumen, which provides a nutrient-rich environment for microbes such as *O. formigenes* to (re)colonize and degrade oxalate prior to absorption

into the host circulation. A recent study (*Canales and Hatch, 2017*) of a surgery-induced EH rat model via Roux-en-Y gastric bypass (RYGB), indicated the potential of *O. formigenes* for treating EH in the post-RYGB rat, *O. formigenes* strain OXWR achieved 100% colonization and decreased urinary oxalate by 74% compared to 39% in the sham-operated group. However, levels of *O. formigenes* colonization in EH patients need to be established in future studies. Although PH is a rare disease, EH in IBD patients is common and growing more so, with currently limited treatment options.

Although our study has significant findings with the potential for translational and mechanistic studies, we acknowledge study limitations. The fact that we only used sequences of proteins instead of the whole genomes during read mapping could lead to false taxa assignments due to possible horizontal gene transfer events. Our analysis was limited to the currently characterized ODPs; therefore, we cannot rule out the existence of other enzymes in the human microbiota to degrade oxalate. Our observations largely rely on metatranscriptomics data, and therefore could be limited by technical biases. It is more difficult to acquire high-quality metatranscriptome, as RNA is less stable and subject to degradation during sample preparation. However, the high consistency across the separate studies (*Figure 2*, *Figure 2—figure supplement 3*, *Figure 3—figure supplement 1*) done using different sample collection methods, library preparation, and sequencing methods (*Supplementary file 1a*), suggests that our findings are robust with reference to technical variation.

Also, we did not have access to urinary oxalate or ascertain kidney stone history in our IBD cohort, both critical variables to extend our conclusions. However, previous studies demonstrated intestinal oxalate is predominantly absorbed paracellularly (*Binder, 1974*; *Knauf et al., 2011*; *Saunders et al., 1975*), and oxalate absorption is determined by concentration gradient, gut permeability, and oxalate bioavailability. Thus, colonic oxalate levels should correlate with both renal oxalate and with risk for CaOx stones. In addition to the untargeted metabolomics data used in this study, targeted measurement of fecal and urinary oxalate is desired in future controlled human studies. Oxalate transport in the human intestine has not been completely elucidated, nor has the relative importance of the small and large intestine to oxalate absorption been affirmatively determined. Therefore, colonization site and biogeography may be important factors for oxalate-degrading microbes to reduce host oxalate absorption. The ODP-transcribing microbes identified in the present study are based on fecal samples, which may bias towards organisms colonizing the colon, but not the small intestine. Furthermore, we showed in mice that *O. formigenes* colonization significantly affects oxalate homeostasis in vivo but further colonization with synthetic communities of different oxalate degraders is needed to identify which oxalate-degrading species have the largest impact on oxalate degradation and overall oxalate homeostasis.

In prior human studies, *O. formigenes* alone can effectively decrease host fecal or urinary oxalate levels, when host are exposed to high oxalate levels (*Canales and Hatch, 2017*; *Jiang et al., 2011*; *Li et al., 2016*; *Li et al., 2015*). In our study, *O. formigenes* colonization was below the level of detection in the jejunum and ileum; therefore, it is likely that *O. formigenes* predominately colonizes and performs oxalate degradation in the colon, reducing dietary oxalate being absorbed into the host. A study with germ-free mice (*Li et al., 2016*) observed that colonization of *O. formigenes* strain OXCC13 decreased the mouse fecal oxalate, which is consistent with our findings, but not urinary oxalate. However, several differences were noted between their study and ours, most importantly the use of different mouse strain and their use of a lower oxalate and higher calcium diet (0.1% oxalate and 1% calcium). In our study, both *O. formigenes* strains we tested reduced host urinary and fecal oxalate, although PH1 that was isolated from human showed a greater effect. This difference suggests that establishing the metabolic activity and host adaptability is critical for evaluating the therapeutic potential of individual *O. formigenes* strains.

## Materials and methods

### Meta-omics data of the human microbiome

Metagenomic and metatranscriptomic data of healthy human subjects were collected from five and four studies, respectively (*Abu-Ali et al., 2018*; *Ehrlich, 2011*; *Le Chatelier et al., 2013 Franzosa et al., 2014*; *Lloyd-Price et al., 2019*; *Petersen et al., 2017*; *Schirmer et al., 2018*). Metagenomic and metatranscriptomic data of healthy humans and IBD subjects were collected from the iHMP-IBD study (*Lloyd-Price et al., 2019*). Each sample was cleaned by KneaData to remove

low-quality reads and host-associated reads. The metabolic profiles of each sample were surveyed using HUMAnN2 v0.11.1 (*Franzosa et al., 2018*) under parameters `-prescreen-threshold` *0.01*, `-pathways-database metacyc_reactions_level4`, `metacyc_pathways_structured`, and `-protein-database` *uniref50*, for the comparison in *Figure 2—figure supplement 1*. Fecal oxalate relative abundance was determined from untargeted metabolomics data from iHMP-IBD; measurements related to oxalate were selected and normalized against total metabolite (percent abundance of all observed metabolites) for analysis.

## Homologous proteins of ODE

The protein homolog families of OXDD, FRC, and OXC were characterized by UniProt Interpro (*Mitchell et al., 2019*; *Mulder et al., 2005*) (V70) in protein families IPR017774, IPR017659, and IPR017660, respectively. We acquired the taxonomic origin and amino acid sequences of 2699 OXDD, 1947 FRC, and 1284 OXC homologs. Protein homologs that are 100% identical were then removed, leaving 2519 OXDD, 1556 FRC, and 1037 OXC unique homologs, which were used as a reference database of ODEs for subsequent query against the meta'omics data. Oxalate oxidoreductase (*Figure 1A*), a recently discovered enzyme for which there only is limited information (*Anand et al., 2002*; *Dumas et al., 1993*; *Grąz et al., 2016*; *Kumar et al., 2011*; *Svedružić et al., 2007*; *Tanner et al., 2001*), was not considered in this present study.

## Pairwise identity between ODE protein homologs

Multiple sequence alignments were performed among the unique protein homologs separately, by muscle (*Edgar, 2004*) in seaview v4.7 (*Gouy et al., 2010*), and alignments were trimmed and imported into R. The pairwise alignment distance $d$ was calculated using function *dist.alignment* in the seqinR package (*Charif et al., 2005*) based on identity or Fitch matrix (*Fitch, 1966*). The alignment distance $d$ was subsequently converted to percent protein identity $100 * (1 - d^2)$, following the documentation of *dist.alignment*.

## Detection of ODE in the meta-omics data

We used two different approaches with complementary methodologies to ensure the conclusions are robust.

1. The first method we used is an agnostic approach that uses full protein sequences with high sensitivity. The quality-filtered meta'omics data were aligned against the reference protein databases consisting of the unique ODE homologs, by diamond blastx (*Buchfink et al., 2015*), with best hit returned (–max-target-seqs 1). Alignments with identity <90% were arbitrarily filtered out to prevent non-ODE reads from aligning to the ODE-specific reference proteins due to local similarity (*Figure 2—figure supplement 2*) By setting a stringent identity cutoff of 90%, we retain only the high-confidence alignment pairs. The abundance of each ODE protein homolog was calculated as reads per kilobase per million (RPKM) in each sample. When multiple timepoints were available, each subject was represented by the mean measurements across all samples provided.

2. We also used ShortBRED, which compares ODEs with all other known proteins to identify highly specific marker peptides, and thus can achieve high specificity. Using *ShortBRED-identify*, FRC, OXC, and OXDD were clustered into 202, 190, and 846 families respectively, with a centroid/representative protein selected for each family (*Supplementary file 2*). Then those centroids were compared against the uniref90 reference protein database (the ODE homologs were excluded), to remove the common region for identification of short peptide markers (*Supplementary file 3*). Lastly, filtered meta-omics reads were mapped against those peptide markers using *ShortBRED-quantify* with parameters `-pctlength` *0.5* and `-id` *0.9*, to calculate the RPKM for each protein family. Default parameters were used unless noted otherwise.

## Population-level contribution to ODE

The population-level contribution of a species to ODE was designed as a measurement to take both prevalence and abundance information into consideration. It is calculated for each ODE separately, based on their abundances (RPKM values). Using oxc as the example, suppose there are $M$ oxc-coding species and $N$ samples. In any given sample $j$, the contribution of species $i$ to OXC, $c_{ij}$, is represented by its relative oxc abundance, calculated from

$$c_{ij} = z_{ij} / \sum_{i=1}^{M} z_{ij}$$

where the $z_{ij}$ denotes the RPKM$_{oxc}$ of species $i$ in sample $j$. In this way, we normalize across samples with the total contribution in any OXC-positive samples fixed to 1, and to 0 in any OXC-negative samples.

The population-level contribution of species $i$: $C_i$, can be subsequently calculated from summating contribution of species $i$ in $N$ samples, as follows

$$C_i = \sum_{j=1}^{N} c_{ij}$$

Note that population-level contribution of species monotonically increases with sample size $N$. Therefore, it has been transformed to relative scale when being compared across different populations or different sample types (metagenome vs. metatranscriptome), such as in *Figure 3C*, *Figure 3—figure supplement 4* and 12.

## BURRITO for linking function to taxa

BURRITO (*McNally et al., 2018*) was used to deconvolve ODE genes into taxa at the species level. Taxonomic profiles of healthy metagenome (n = 2539 samples) were generated by Metaphlan2 (*Truong et al., 2015*) under default parameters. In each sample, RPKM of all FRC or OXC homologs were summed as total abundance of *frc* or *oxc* genes and then supplied to BURRITO. The genomic content file was derived from taxonomic annotation of protein homologs from UniProt Interpro. We assumed one copy of *frc* and *oxc* for each genome to minimize bias, as such information is not available for all species.

## Network analysis

A network analysis of *oxc* or *frc* expression from microbial species used SpiecEasi (*Kurtz et al., 2015*). The raw RPKM values were used, and networks were constructed under default parameters *method='mb'*, *sel.criterion='bstars'*, *lambda.min.ratio = 2e-2*, *nlambda = 100*, and *pulsar.params=list(rep.num = 20, ncores = 2)*.

## Code availability

Source code of the pipeline can be found on Github via https://github.com/ml3958/FindTaxaCtrbt (*Liu, 2021*; copy archived at swh:1:rev:13bbc4662f458bff327e348162bf51-d875ed34d3). Downstream analysis scripts are available per request.

## *O. formigenes* isolation and culturing

*O. formigenes* strain PH1 was isolated from the stool sample of a primary hyperoxaluria patient using methods described in *Allison et al., 1985*. Both strains were cultured in defined oxalate broth as described (*Allison et al., 1985*), except that the concentration of oxalate is 50 mM. *O. formigenes* was cultured at 37°C in an anaerobic chamber before each mouse gavage.

## *O. formigenes* colonization of mice

A total of 17 C57Bl/6 male mice maintained in our gnotobiotic facility were used. The median of mouse baseline microbiome 16S qPCR measurements was $2.6 \times 10^5$, which is significantly lower than that of SPF mice (usually $10^8$ or $10^9$ in our experiments, representing <0.1% of the absolute abundance). The taxa detected in the baseline samples (*Supplementary file 1e*) did not lead to colonization resistance to the introduced *O. formigenes* strains.

At days 0, 2, and 7, mice were gavaged with 100 µl from a 24 hr growing culture of *O. formigenes* strain OXCC13 (n = 5), *O. formigenes* freshly isolated from a PH1 subject (n = 6), and *O. formigenes* culture medium alone (n = 6). Mice were fed normal chow supplemented with 1% sodium oxalate from study commencement (day −5) until mouse sacrifice at day 35. We did not perform an explicit power analysis since there is no previous data on our animal model and experimental conditions. We thought that five to six mice per group is a reasonable number of mice per group to show a

significant change in urinary oxalate. Biological replicates are parallel measurements of biologically distinct samples that capture random biological variation, which may itself be a subject of study or a noise source. In each mouse group, we performed five to six biological replicates by gavaging material (culture media, growing culture of strain OXCC13 or PH1) into five to six mice.

## DNA extraction and *O. formigenes* quantitative PCR

Murine fecal pellets or intestinal contents were collected and stored at −80°C until DNA was extracted, using the MoBio 96-well extraction kit, following the manufacturer's instructions. We confirmed that mice were colonized with *O. formigenes* at sacrifice using qPCR of fecal samples (Figure S16). qPCR was used to quantitate the number of copies of the oxc mRNA using the LightCycler 480 SYBR Green I Master Mix and run using the LightCycler 480 system. Paired primers (forward 5′-TGT-TTG-TCG-GCA-TTC-CTA-TC-3′, reverse 5′-TTG-GGA-AGC-AGT-TGG-TGG-3′) were used under the conditions: 95°C for 10 min, followed by 40 cycles of: 95°C for 23 s, 63°C for 20 s, 70°C for 40 s, and a final 30 s at 40°C as described (*Pebenito et al., 2019*).

## Fecal and urinary collections and oxalate measurements

Mice were housed (n = 1 or 2) in metabolic cages, and 48 hr collections (1–2 mice per urine pool) were made under mineral oil into vessels containing crystal thymol as a preservative. Urinary oxalate (mg/dl) and creatinine (mg/dl) concentrations were determined in acidified (HCl) samples collected from all mice over a 48 hr period by Litholink Corp (Chicago, IL). Fecal pellets were collected at the end of the urine collections. Fecal pellets were acidified using 2M HCl, vortexed for 20 min, and then centrifuged at 21,000 g at room temperature, and supernatant fecal water collected using described methods (*Jiang et al., 2011*). Fecal water oxalate was measured using an oxalate calorimetric assay (Abcam, ab196990, Cambridge, UK) per the manufacturer's instructions. Technical replicates are repeated measurements of the same sample that represent independent measures of the random noise associated with protocols or equipment. Fecal oxalate and qPCR were measured in duplicates.

## Acknowledgements

We thank Dr. David Goldfarb and Xuhui Zheng for their helpful comments. We thank Dr. Tim Borbet for the helpful discussion and input. Funding: This study was supported in part by U01AI22285, R01DK110014, and the Rare Kidney Stone Consortium (U54 DK083908) from the National Institutes of Health, by the C and D and Zlinkoff Funds, Oxalosis and Hyperoxaluria Foundation-American Society of Nephrology career development grant, and the TransAtlantic Partnership of the Fondation LeDucq.

## Additional information

### Competing interests

John R Asplin: is an employee of Litholink. Allyson Byrd: is an employee of Genentech. The other authors declare that no competing interests exist.

### Funding

| Funder | Grant reference number | Author |
| --- | --- | --- |
| National Institute of Allergy and Infectious Diseases (NIAID) | U01AI22285 | Martin J Blaser |
| National Institute of Diabetes and Digestive and Kidney Diseases (NIDDK) | R01DK110014 | Huilin Li |
| Rare Kidney Stone Consortium | U54 DK083908 | Lama Nazzal |
| The C & D and Zlinkoff Funds | | Martin J Blaser |
| Oxalosis and Hyperoxaluria | career development grant | Lama Nazzal |

Foundation

| | |
|---|---|
| TransAtlantic Partnership of the Fondation LeDucq | Martin J Blaser |

The funders had no role in study design, data collection and interpretation, or the decision to submit the work for publication.

### Author contributions

Menghan Liu, Conceptualization, Data curation, Software, Formal analysis, Investigation, Visualization, Methodology, Writing - original draft, Writing - review and editing; Joseph C Devlin, Angelina Volkova, Thomas W Battaglia, Melody Ho, Data curation; Jiyuan Hu, Huilin Li, Kelly V Ruggles, Methodology, Writing - review and editing; John R Asplin, Data curation, Writing - review and editing; Allyson Byrd, P'ng Loke, Martin J Blaser, Conceptualization, Resources, Supervision, Funding acquisition, Investigation, Writing - review and editing; Aristotelis Tsirigos, Lama Nazzal, Conceptualization, Resources, Data curation, Supervision, Funding acquisition, Investigation, Methodology, Writing - original draft, Writing - review and editing

### Author ORCIDs

Menghan Liu (iD) https://orcid.org/0000-0002-9390-8194
P'ng Loke (iD) https://orcid.org/0000-0002-6211-3292
Kelly V Ruggles (iD) https://orcid.org/0000-0002-0152-0863
Martin J Blaser (iD) https://orcid.org/0000-0003-2447-2443
Lama Nazzal (iD) https://orcid.org/0000-0003-0106-5060

### Ethics

Animal experimentation: This study was performed in strict accordance with the recommendations in the Guide for the Care and Use of Laboratory Animals of the National Institutes of Health. All of the animals were handled according to approved institutional animal care and use committee (IACUC) protocols (#IA16-00822) of the New York University Langone Medical Center.

### Decision letter and Author response

Decision letter https://doi.org/10.7554/eLife.63642.sa1
Author response https://doi.org/10.7554/eLife.63642.sa2

## Additional files

### Supplementary files

• Supplementary file 1. Tables. (**a**) Description of populations used in the present study. (**b**) Population-level contribution of species to metagenomic or metatranscriptomic OXC, in healthy, ulcerative colitis (UC), or Crohn's disease (CD) patients. (**c**) Population-level contribution of species to metagenomic or metatranscriptomic FRC, in healthy, UC, or CD patients. (**d**) Taxonomic contributions to *frc* or *oxc* genes* inferred by BURRITO in 2359 metagenomic samples (see Materials and methods for detailed description). (**e**) The dominant taxa detected by 16S rRNA sequencing in the baseline fecal samples (n = 17) from the mouse study in *Figure 6*.

• Supplementary file 2. Proteins associated with each protein cluster based on ShortBRED.

• Supplementary file 3. Marker pepetide picked for each protein cluster based on ShortBRED.

• Transparent reporting form

### Data availability

All data generated or analysed during this study are included in the manuscript and supporting files. Source data files have been provided for Figures 2–5.

The following previously published datasets were used:

| Author(s) | Year | Dataset title | Dataset URL | Database and Identifier |
|---|---|---|---|---|
| Petersen LM, Bautista EJ, Nguyen H, Hanson BM, Chen L, Lek SH, Sodergren E, Weinstock GM | 2017 | Athlete Microbiome Project (AMP) | https://www.ncbi.nlm.nih.gov/bioproject/PRJNA305507 | NCBI BioProject, PRJNA305507 |
| Eric FA, Morgan XC, Segata N, Waldron L, Reyes J, Earl AM, Giannoukos G | 2014 | fran | https://www.ncbi.nlm.nih.gov/bioproject/PRJNA188481 | NCBI BioProject, PRJNA188481 |
| Lloyd-Price J, Arze C, Ananthakrishnan AN, Schirmer M, Avila-Pacheco J, Poon TW, Andrews E, Ajami NJ, Bonham KS, Brislawn CJ, Casero D | 2019 | iHMP | https://www.ncbi.nlm.nih.gov/geo/query/acc.cgi?acc=GSE111889 | NCBI Gene Expression Omnibus, GSE111889 |
| Mehta RS, Abu-Ali GS, Drew DA, Lloyd-Price J, Subramanian A, Lochhead P, Joshi AD, Ivey KL, Khalili H, Brown GT, DuLong C, Song M, Nguyen L, Mallick H, Rimm EB, Izard J, Huttenhower C, Chan AT | 2018 | US men | https://www.ncbi.nlm.nih.gov/bioproject/354235 | NCBI BioProject, PRJNA354235 |
| Le Chatelier E, Nielsen T, Qin J, Prifti E, Hildebrand F, Falony G, Almeida M, Arumugam M, Batto JM, Kennedy S, Leonard P, Li J, Burgdorf K, Grarup N, Jørgensen T, Brandslund I, Nielsen HB, Juncker AS, Bertalan M, Levenez F, Pons N, Rasmussen S, Sunagawa S, Tap J, | 2010 | MetaHIT | https://www.ebi.ac.uk/ena/browser/view/PRJEB4336 | EBI European Nucleotide Archive, PRJEB4336 |

Tims S,
Zoetendal EG,
Brunak S,
Clément K,
Doré J,
Kleerebezem M,
Kristiansen K,
Renault P,
Sicheritz-Ponten T,
de Vos WM,
Zucker JD,
Raes J,
Hansen T,
MetaHIT consortium,
Bork P,
Wang J,
Ehrlich SD,
Pedersen O

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
