## [Decision Letter]

**Acceptance summary:**

Oxalate is critical for kidney stones yet the bacteria responsible for its metabolism in humans remain poorly understood. Herein, the authors use a multi-disciplinary approach to study the abundance and expression of genes for human gut bacterial oxalate metabolism in healthy subjects and patients with inflammatory bowel disease. They go on to show that *Oxalobacter formigenes* significantly alters oxalate levels in mice. These analyses provide a critical step towards a more comprehensive view of oxalate metabolism and its role in health and disease.

**Decision letter after peer review:**

Thank you for submitting your article "Microbial contributions to oxalate metabolism in health and disease" for consideration by *eLife*. Your article has been reviewed by 3 peer reviewers, and the evaluation has been overseen by a Reviewing Editor and Wendy Garrett as the Senior Editor. The following individual involved in review of your submission have agreed to reveal their identity: Eric Brown (Reviewer #3).

The reviewers have discussed the reviews with one another and the Reviewing Editor has drafted this decision to help you prepare a revised submission.

Summary:

Liu and colleagues present a series of finding related to gut bacterial oxalate metabolism. First, they curate a set of previously described proteins and generate a reference database of homologs based upon Interpro annotations. Then, they re-analyze previously published meta-genomes and transcriptomes to find hits to these reference genes in the gut microbiomes of healthy and IBD subjects. Surprisingly, they find an inverse association between oxalate levels and the total transcripts of bacterial oxalate degradation genes. They also include data showing that *Oxalobacter formigenes* (a model oxalate degrading bacterium) impacts oxalate in mice. These analyses are a good example of how the microbiome field as a whole can utilize complex multi-omic datasets to help answer specific questions of clinical importance. However, there are multiple limitations and points which need to be addressed prior to publication.

Essential revisions

1. The claim that *O. formigenes* is the dominant oxalate degrading species is not well supported by any of the current data. The sequence analysis is based on a presumably partial knowledge of the full scope of enzymes capable of this activity, so it remains unclear if alternative species or pathways are important to consider. The mouse experiment is used as a "validation" but only shows that this species is sufficient to impact oxalate not that it is necessary in humans. A valuable first step would be to colonize germ-free mice with *O. formigenes* along with multiple other oxalate degraders, then perform leave-one-out experiments to test which species have a marked impact on oxalate levels when removed.

2. The approaches used to assign genes and species are not state-of-the-art and may not be entirely reliable. I'd suggest trying ShortBRED (Huttenhower lab) or a related tool to quantify the protein families of interest. FishTaco and BURRITO (Borenstein lab) could be used to help link taxonomy to function. This is an important point since it relates to the claim that *O. formigenes* is the source of most transcripts. Furthermore, it's unclear if these genes are horizontally transferred (which could be assessed by comparing gene and species trees). If so, simple read mapping could assign genes to the wrong genomes. Ignoring these other tools, the validation shown in Figure S3 doesn't make much sense to me. The threshold of 90% misses many of the intraspecies comparisons. I'm also concerned that Oxalobacter, the focus of this work, only has a handful of representative genes, which will make it difficult to reliably assign reads to this genus let alone to *O. formigenes* specifically.

3. Some attempt needs to be made to experimentally address the counter-intuitive observation that higher substrate (oxalate) is associated with lower expression, which runs counter to how most bacterial genes are regulated. What accounts for the downregulation? Is this related at all to the environment within the IBD gut?

4. The way the oxalate levels in the feces are presented is problematic. In the manuscript, the authors make multiple mentions the observed abundance of oxalate is a "fecal concentration of oxalate" when in fact it is the relative abundance of oxalate as measured by LC-MS. These data are not measuring concentration but relative abundance, which can be influenced by other non-biological factors such as how well the metabolite is ionized in each sample by LC-MS. Authors should not these are relative abundance calculations and not concentrations (for example Line 13 describing Figure 4A). Furthermore, the authors should indicate whether the samples were normalized between cohorts and how they were, and whether the relative abundance measurement is correct for within sample differences (% abundance of all observed metabolites) or a raw abundance? For example, the large difference between IBD and healthy stool could lead to less total metabolites being extracted and ionized thus data normalization between samples may actually increase the correlations you are seeing between oxalate and oxalate-degrading enzyme expression by meta-transcriptomics.

[Editors' note: further revisions were suggested prior to acceptance, as described below.]

Thank you for submitting your article "Microbial genetic and transcriptional contributions to oxalate degradation by the gut microbiota in health and disease" for consideration by *eLife*. Your article has been reviewed by 2 peer reviewers, including Peter Turnbaugh as the Reviewing Editor and Reviewer #1, and the evaluation has been overseen by Wendy Garrett as the Senior Editor.

Essential revisions:

1. In Figure 4 the raw values in the y-axis when reporting oxalate relative abundance are unclear which units? Are they actually negative. The figure legends in general should include more specific information for the metabolomic data so it is easier to interpret.

2. The authors still refer to "oxalate concentrations" in the figure legend for figure 4.

3. It is unclear why the calprotectin cut-off was 50ug/mL please cite literature or a reason for using this value. It seems like there is in fact a trend with inflammation potentially.

4. Along that note Pearson analysis could show whether calprotectin levels correlate with Oxalobacter, something worth mentioning for the differences in abundance across IBD vs healthy controls and will be useful for the field if in fact this were to ever be utilized as a probiotic by others in the future.

5. More clarification is needed in the manuscript text on how comparing metagenomic analysis with metatranscriptomics can successfully pinpoint which taxa contribute to a disease pathway (in this case oxalate degradation). I still find this comparison confusing as a reader and potential pitfalls to this approach should be more clearly stated (sequencing depth as mentioned).

---

## [Author Response]

Essential revisions1. The claim that O. formigenes is the dominant oxalate degrading species is not well supported by any of the current data. The sequence analysis is based on a presumably partial knowledge of the full scope of enzymes capable of this activity, so it remains unclear if alternative species or pathways are important to consider. The mouse experiment is used as a "validation" but only shows that this species is sufficient to impact oxalate not that it is necessary in humans. A valuable first step would be to colonize germ-free mice with O. formigenes along with multiple other oxalate degraders, then perform leave-one-out experiments to test which species have a marked impact on oxalate levels when removed.

We agree with the reviewer that we do not know the full scope of enzymes capable of this activity, however we based our analysis and conclusions on the published literature, which indicates that oxalate degradation is through one of the 2 pathways (Figure 1). We now acknowledge this limitation in the Discussion.

We changed the description of the animal experiment to “To validate our bioinformatics prediction that *Oxalobacter formigenes* is an important oxalate degrading organism that can influence oxalate homeostasis in vivo, we examined whether the colonization of *O. formigenes* results in significant reduction in urinary oxalate in mice.”

We also thank the reviewer for the suggestion for a ‘leave-one-out’ experiment, which is in our future plans to test the hypothesis suggested by the data. We agree with the reviewer that an experiment in which we could give GF mice a synthetic community of potential oxalate-degrading microbes, and monitor their in vivo colonization, ODP transcription, and host oxalate metabolic responses will be valuable. However, such an experiment would require careful design and multiple pilot experiments. We would first need to test different diets of varying oxalate levels to establish a mouse model with intra-colonic oxalate concentrations faithfully mimicking the human intra-colonic milieu and avoiding toxicity to the oxalate-degrading community with high concentrations of dietary oxalate, as has been described (Miller, Dale et al. 2017). We also need to test multiple strains from the same species of oxalate degraders to ensure rigor. Thus, in combination with the current COVID situation limiting our capabilities to perform further animal experiments, we would like to argue that this current experiment is beyond the scope of this manuscript and propose to perform it when possible in the future, as part of a future manuscript. We have now added this limitation to the Discussion and the need for validating our results using animal models colonized with these different microbiota. We believe that the present manuscript has a number of important observations, not requiring this question to be answered, for it to still substantially advance the field.

2. The approaches used to assign genes and species are not state-of-the-art and may not be entirely reliable. The approaches used to assign genes and species are not state-of-the-art and may not be entirely reliable. I'd suggest trying ShortBRED (Huttenhower lab) or a related tool to quantify the protein families of interest.

As the reviewer suggested, we reanalyzed abundances of oxalate degrading enzymes (ODEs) using ShortBRED. First, using *ShortBRED-identify* default parameters, we clustered FRC, OXC, and OXDD homologs included in this study into 202, 190, and 846 families, respectively. Marker peptides were identified for each of the ODE protein families (see Methods). Next, using *ShortBRED-quantify*, we mapped the multi-omics reads of 2359 metagenomic and 1053 transcriptomic samples to those marker peptides and quantified abundances of each ODE using *ShortBRED-quantify*.

The results generated by these new *ShortBRED* analyses validated all of our prior major findings. We found that:

1. FRC and OXC, but not OXDD, are frequently detected in the human gut metagenome and metatranscriptome (New Figure 3—figure supplement 5A), which is consistent with our previous observation (Figure 3).

*2. O. formigenes* is the species with the highest transcriptional activity for FRC (New Figure 3—figure supplement 5B). Specifically, each of the three *O. formigenes* FRC homologs (C3X9Y2, C3X762, and C3X2D4) are distinct from other homologs and from each other. Thus each formed a singleton family (New Supplementary file 2) with unique peptide markers (New Supplementary file 3). Based on the ShortBRED analysis, the three *O. formigenes* FRCs are the most commonly transcribed among the FRCs encoded by any taxon. They are present in the metatranscriptome of 50, 52, and 41 percent of the tested subjects, respectively (New Figure 3—figure supplement 5B, right), and also are among the most abundant ones (rank 1,2 and 5) among all FRC proteins (New Figure 3—figure supplement 5B, right). Therefore, *O. formigenes* is the dominant microbial source for FRC in the human microbiome (See below for OXC data).

3. Based on *ShorBRED-quantify* results, the global transcripts for *frc* and *oxc* are reduced in all IBD groups compared to healthy individuals, particularly in UC and CD-L3 IBD patients (New Figure 3—figure supplement 6), which is consistent with our previous observation (Figure 4).

However, ShortBRED relies on many arbitrary decisions (including CD-Hit, centroid protein selection, length and identity of short identical regions and final marker). It is not possible to empirically determine the best parameter combinations. For example, ShortBRED did not detect *O. formigenes* OXC in any sample, which contradicts all of our other analyses using ShortBRED and our method (Figure 3 and Figure 3—figure supplement 3), and contradicts the well-established fact that *O. formigenes* colonizes >30% of humans (Kelly, Curhan et al. 2011, PeBenito, Nazzal et al. 2019). We reasoned that this under-detection is due to poorly-selected marker peptides. Indeed, the makers selected for the protein family of *O. formigenes* OXC (New Figure 3—figure supplement 6A, markers 1-20) are (1), short (median length 16.2 amino acids, as a comparison: the marker for *O. formigenes* FRC is 300 amino acids), and (2), improperly broken up from long continuous regions (New Figure 3—figure supplement 6A). Therefore, we manually generated new markers (Markers #21-24) by combining markers that were one amino acid apart. Using these longer markers, ShortBRED now successfully detected OXC using 3 of the 4 markers (except for the shortest marker) with the expected patterns (New Figure 3—figure supplement 6B, C).

ShortBRED and our previous method are different and both have unique strengths. Our previous method uses protein full length information and is completely agnostic, relying on best alignment hit to assign read to reference. Our method successfully detected *O. formigenes* FRC and OXC (Figure 3 and Figure 3—figure supplement 3). Thus, the two methods have different advantages and are complementary.

In summary, using a different methodology, several of our major conclusions were confirmed. We now report the results in the revised manuscript in the Results section *Validation of ODP detection using ShortBRED* and in New Figure 3—figure supplement 5, Figure 3—figure supplement 6. We also discuss the pros and cons of the two methods to help readers interpret the data.

FishTaco and BURRITO (Borenstein lab) could be used to help link taxonomy to function.

Thank you for this excellent suggestion. We now used BURRITO (McNally, Eng et al. 2018) to link function to taxa based on the taxonomic composition generated by Metaphlan2 (Truong, Franzosa et al. 2015) (See Methods). Only metagenomes (n=2359 samples) were included for this analysis because the method was designed to be used with metagenomic data in which the taxonomic and gene abundance are linearly linked.

The taxonomic contributions based on BURRITO (New Supplementary file 1d) are highly concordant with the findings using our original method (Supplementary file 1b, metagenome, Healthy). *E. coli* and *O. formigenes* are consistently the top two contributors, and several *Bifidobacterium* spp and *Lactobacillus* spp contributed to a lesser extent. We now added those new results into the revised manuscript.

Furthermore, it's unclear if these genes are horizontally transferred (which could be assessed by comparing gene and species trees). If so, simple read mapping could assign genes to the wrong genomes.

The reviewers’ concern regarding horizontal gene transfer (HGT) is well-taken. FRC and OXC homologs encoded by the same Class of bacteria generally cluster together (Author response image 1), suggesting this pathway is evolutionary conserved at high taxonomic levels. As microbiomes are highly individual-specific, to address HGT, one needs to assemble reads from each metagenome to highquality contigs to provide fine resolution to assess for HGT. Currently such a task is technically and computationally challenging. As it is not the focus of current study, we acknowledge these limitations and now include the following sentence in the Discussion:

“The fact that we only used sequences of proteins instead of whole genomes during read mapping could lead to false taxonomic assignments due to possible horizontal gene transfer events”

**Author response image 1. respfig1:** Phylogenetic analysis of OXC (A), FRC (B) uniref100 proteins. Each tip represents a protein homolog, which is color-coded by the microbial Class associated with the encoding. The tip size is proportional to the prevalence of the corresponding protein in the metatranscriptomes of 165 healthy individuals. Homologs with prevalence >1% are annotatedwith text.

I'm also concerned that Oxalobacter, the focus of this work, only has a handful of representative genes, which will make it difficult to reliably assign reads to this genus let alone to O. formigenes specifically.

We thank the reviewer for raising this concern. First, *Oxalobacter* is currently a small genus consisting of only two known species (The other species *Oxalobacter vibrioformis* was isolated from anoxic freshwater sediment and only described once in 1989 (Dehning and Schink 1989). Thus, we are confident that both *Oxalobacter* genus and *O. formigenes* species are highly-identifiable taxa. Second, the *frc* and *oxc* genes of *O. formigenes* species are distant from other species, as evidenced by the fact that ShortBRED CD Hit clustered them as individual families at the amino acid level) (New Supplementary file 2). Especially for FRC, all sequenced *O. formigenes* genomes encode three copies of the *frc* gene with conserved operon structures (Author response image 2). The three *frc* genes are further grouped into two forms (Author response image 2), both of which were identified in the current study. Collectively, the *O. formigenes frc* and *oxc* genes in our study are highly representative, and the read assignments for them should be highly specific as well.

**Author response image 2. respfig2:** Operon structure and phylogenetic relationship of FRCs of four *O.formigenes* strains OXCC13, HC1, HOxBLS, and OXK, for which whole genome sequence is available. (A). Schematic representation of the operon structures for the three FRC genes. The structures are conserved across all four strains. We named the three FRCs as α-, β-, and γ-FRC (from top to bottom). (B). Maximum likelihood phylogenetic tree for the α-, β-, and γ-FRCs from four *O. formigenes* strains, based on their amino acid sequence. The sequence alignment and tree were generated using phylogeny.fr using “simple click” mode.

The threshold of 90% misses many of the intraspecies comparisons.

We thank the reviewer for this comment. The 90% cutoff on an alignment hit is to prevent non-ODE reads being aligned to our ODE reference proteins at a low identity score. We now clarify this point in the text, as follows:

“By setting a stringent identity cutoff of 90%, we retain only the high confidence alignment pairs. The finding generated using this 90% cutoff is consistent with the observations generated through a second method ShortBRED, which uses ODE-specific marker peptides, which provide evidence that the cutoff is highly effective at preventing spurious alignment.”

3. Some attempt needs to be made to experimentally address the counter-intuitive observation that higher substrate (oxalate) is associated with lower expression, which runs counter to how most bacterial genes are regulated. What accounts for the downregulation?

The Reviewer is correct, and we thank him/her for the comment. The ‘downregulation’ of *oxc* or *frc* expression by microbiome referred to the reduction of total transcripts associated with *oxc* and *frc*. The suppression or absence of *O. formigenes* cells is largely responsible for the decrease in ODP transcripts in the IBD gut (Figure 5A). To avoid confusion, we have revised our manuscript, replacing “downregulation of ODP” with “global reduction of ODP-related transcripts” throughout.

Is this related at all to the environment within the IBD gut?

IBD patients regularly receive antibiotics, and *O. formigenes* is susceptible to commonly used antibiotics (Mittal, Kumar et al. 2005, Kharlamb, Schelker et al. 2011, Lange, Wood et al. 2012, Liu, Koh et al. 2017). Our unpublished human data in healthy adults (in revision, 2021) demonstrated that a single course of antibiotics (clarithromycin and metronidazole) to eradicate *Helicobacter pylori* results in the persistent suppression of *O. formigenes* colonization for at least 6 months. Other changes in the intestinal milieu of IBD patients affecting pH, oxygen levels, and increasing bile acids, have been shown to impact *O. formigenes* in vitro (Allison, Dawson et al. 1985, Duncan, Richardson et al. 2002). We have now added these potential explanations into the Discussion.

4. The way the oxalate levels in the feces are presented is problematic. In the manuscript, the authors make multiple mentions the observed abundance of oxalate is a "fecal concentration of oxalate" when in fact it is the relative abundance of oxalate as measured by LC-MS. These data are not measuring concentration but relative abundance, which can be influenced by other non-biological factors such as how well the metabolite is ionized in each sample by LC-MS. Authors should not these are relative abundance calculations and not concentrations (for example Line 13 describing Figure 4A). Furthermore, the authors should indicate whether the samples were normalized between cohorts and how they were, and whether the relative abundance measurement is correct for within sample differences (% abundance of all observed metabolites) or a raw abundance? For example, the large difference between IBD and healthy stool could lead to less total metabolites being extracted and ionized thus data normalization between samples may actually increase the correlations you are seeing between oxalate and oxalate-degrading enzyme expression by meta-transcriptomics.

We thank the Reviewer for these helpful comments. We address the Reviewer’s comment point-by-point below:

Authors should note these are relative abundance calculations and not concentrations (for example Line 13 describing Figure 4A). Furthermore, the authors should indicate whether the samples were normalized between cohorts and how they were, and whether the relative abundance measurement is correct for within sample differences (% abundance of all observed metabolites) or a raw abundance?

We thank the reviewer for this comment. As suggested, we have repeated all analyses based on the relative abundance of oxalate (% abundance of all observed metabolites in each sample), to account for sample differences (New Figure 4, which replaces the prior Figure 4). This point now is indicated in the Methods under section Meta-omics data of the human microbiome.

Using relative abundance, we reached the same observation that fecal oxalate is elevated in all IBD cohorts compared to healthy individuals (New Figure 4). The correlation between *oxc* expression and oxalate relative abundance showed negative, but not statistically significant, trends in all IBD groups (New Figure 4—figure supplement 4A), but *frc* was significantly inversely correlated with oxalate relative abundance in the UC group (New Figure 4—figure supplement 4A). In the revised manuscript, we now report the results based on both absolute and relative abundances.

[Editors' note: further revisions were suggested prior to acceptance, as described below.]

Essential revisions:1. In Figure 4 the raw values in the y-axis when reporting oxalate relative abundance are unclear which units? Are they actually negative. The figure legends in general should include more specific information for the metabolomic data so it is easier to interpret.

As requested by a prior reviewer (Prior Review point 4), Figure 4A reports log_10_ relative abundance of oxalate as a percent of the total metabolites measured (See Methods). The negative values are due to log_10_ transformations as indicated in the figure. To improve clarity, we now include that information in the figure legend as follows:

“(A). Stool oxalate relative abundance (log_10_) in healthy, UC, CD, CD-L3 or CD-nonL3 subjects from the HMP-IBD study. Fecal oxalate relative abundance was determined from untargeted metabolomics data from iHMP-IBD; measurements related to oxalate were normalized against total metabolites (percent abundance of all observed metabolites) for analysis.”

2. The authors still refer to "oxalate concentrations" in the figure legend for figure 4.

We thank the reviewer for catching this error. We now have updated the figure axis and legend of Figure 4 and the related Supplemental Figures 1, 2, 4, and 5 to indicate “Fecal oxalate log_10_ relative abundance”.

3. It is unclear why the calprotectin cut-off was 50ug/mL please cite literature or a reason for using this value. It seems like there is in fact a trend with inflammation potentially.

The fecal calprotectin cutoff was determined based on the pertinent literature (Damms and Bischoff, 2008; Manz et al., 2012; Pathirana, Chubb, Gillett, and Vasikaran, 2018) and to be consistent with current test kits (Pathirana et al., 2018). We have now included citations to these references in the main text and in the Figure Legend.

As the Editor/reviewer points out, increased fecal oxalate with higher inflammation levels (fecal calprotectin >50µg/g) is indeed present. A sub-analysis with samples with fecal calprotectin > 50ug/ml showed a positive correlation between fecal calprotectin and oxalate (new Figure 4—figure supplement 2C), which needs to be examined with larger patient cohorts in future studies. This point now is included in the text, as “We observed increased fecal oxalate in association with higher inflammation levels (fecal calprotectin >50µg/ml) (Figure 4—figure supplement 2C); since our number of study subjects was small, this relationshp needs to be examined with larger patient cohorts in future studies.”

4. Along that note Pearson analysis could show whether calprotectin levels correlate with Oxalobacter, something worth mentioning for the differences in abundance across IBD vs healthy controls and will be useful for the field if in fact this were to ever be utilized as a probiotic by others in the future.

We agree, thank you. Using detection of *O. formigenes* ODP genes and transcripts jointly as a marker for its presence, we observed significantly elevated fecal calprotectin levels when *O. formigenes* is absent in healthy individuals and in CD-L3 patients. This analysis suggests that colonization by *O. formigenes* and gut inflammation could potentially be linked. As suggested by the Editor, we now include new Supplemental Figure 2D, which shows the relationship of *O. formigenes* status and fecal calprotectin level, without using an arbitrary cut-off. Although we show this Figure, we indicate that the underlying mechanism is unknown and needs to be examined in future studies. The new text that we have included is as follows: “Significantly elevated fecal calprotectin levels were observed when *O. formigenes* was absent in healthy individuals and in CD-L3 patients (Figure 4—figure supplement 2D) “

5. More clarification is needed in the manuscript text on how comparing metagenomic analysis with metatranscriptomics can successfully pinpoint which taxa contribute to a disease pathway (in this case oxalate degradation). I still find this comparison confusing as a reader and potential pitfalls to this approach should be more clearly stated (sequencing depth as mentioned).

We appreciate the reviewer’s suggestion. We now more clearly describe the two critical advantages of an approach that co-analyzes metagenome and metatranscriptome in the discussion, as follows:

“(1) Detection of genes in the metagenome does not ensure that they are being actively utilized by the indicated taxon. Thus, analyzing metatranscriptomic data allows distinguishing the taxa that are actively contributing to a biological process by generating the relevant proteins. In contrast, for those taxa that are not, the analysis provides a new view of their metabolism, in that although they have the potential for using a certain pathway or metabolite, it is not active under the particular condition tested. (2) Having metatranscriptomic data or other functional readout is particularly useful for cross-comparisons of the microbiome of different cohorts (i.e., diseased versus healthy subjects, in humans or experimental animals). For example, in our study, ODP genes were increased in IBD patients, whereas transcripts were significantly reduced. Specifically, the increased ODP gene abundances were due to the over-representation in IBD patients of *E. coli* strains that uncommonly transcribe ODP in vivo. In contrast, the decrease of oxalate-degrading gene expression is caused by the loss of *O. formigenes*, which is the dominant microbe that transcribes this pathway. Having observed the differential abundance for a gene does not necessarily indicate functional shifts, as gene abundance is driven by the most abundant taxa, which could be transcriptionally silent. As such, the transcriptional evidence is a better indicator for evaluation of microbiome functional differences.”

In addition to the previous description of the limitations of our methods, we now have added another limitation that “it is more difficult to acquire high-quality metatranscriptome, as RNA is less stable and subject to degradation during sample preparation”.

[Editors' note: we include below the reviews that the authors received from another journal, along with the authors’ responses.]

Reviewer 1The authors have taken advantage of accrued data associated with the Microbiome Project to identify gut microbes in humans that are involved in the breakdown of oxalate. They importantly and for the first time use both metagenomics and metatranscriptomics to better understand microbial contributions to oxalate degradation in the human gut. They also performed analyses in various types of IBD to determine if these patients have perturbations of the oxalate degrading microbiome. Some novel aspects of the study were (1) classifying the abundance of oxalatedegrading organisms based on the enzymes involved and their co-factor requirements (2) revealing that the contribution of the oxalate auxotroph O. formigenes to the oxalate degrading pathway is greater than the transcriptomic contributions of all other species combined (3) demonstrating that oxalate degradation pathways were reduced and fecal oxalate elevated in patients with IBD compared with healthy controls. This study opens the doorway to the development of strategies that may off-set the increased oxalate absorption associated with IBD.

We appreciate the reviewer’s comment on the multiple novel aspects of the study. We agree that as the first systemic study on oxalate-degradation by human gut microbes, this work will open new avenues of therapeutic strategies for enteric hyperoxaluria (EH). This study for the first time describes the relative importance of different microbial taxa in human oxalate degradation, and identifies the most relevant clinical indication (Crohn’s disease with the ileocolonic phenotype).

Points that should be addressed:1. Figure 4- Indicate if this is a log scale.

Thank you for this suggestion. The fecal oxalate level in panels A and D was on a log_10_ scale, but this was not reflected on the axis title. We now have updated the axis title as “Log_10_ fecal oxalate by LC-MS” (please see new Figure 4 on page 37).

2. Due to the importance of fecal oxalate measurements, the authors should add to their limitations that a more refined and targeted direct measure of fecal oxalate is warranted to substantiate these claims.

Thank you for the suggestion. We have now added a sentence in the Discussion section that states “In addition to the untargeted metabolomics data used in this study, targeted measurement of fecal and urinary oxalate is desired in future controlled human studies.”.

In addition, in a mouse experiment that we now present, both fecal and urine oxalate levels were directly measured using targeted assays (please see the new Figure 6B and Figure 6C on page 39).

3. Figure S10 (B). Legend indicates OXC abundance, but axis title is frc.

Thank you for pointing out this error. We now have updated that figure (see page 54).

Reviewer 2The analysis is overall well done and convincing.

We thank the reviewer for the positive feedback.

1. Inflammasome pathways have been implicated but may not be the only mechanism of renal damage in hyperoxaluria.

Thank you for the comment. We agree with the reviewer that there are other mechanisms of calcium oxalate nephrotoxicity. Due to the word limits, we did not list all relevant studies. Now we have added text to the Introduction about other potential mechanisms of calcium oxalate nephrotoxicity.

2. There is no direct evidence that elevated intestinal oxalate is key to EH, although there us indirect evidence, especially in older literature from the 1980s

We thank the reviewer for this comment. We agree that there is no direct evidence yet linking fecal oxalate to EH. However, in prior EH human studies and animal models, fecal oxalate was not measured. In our study, we observed increased fecal oxalate in ulcerative colitis patients and Crohn’s disease patients with the ileocolonic phenotype, which is the same subgroups as have clinical nephrolithiasis risk (Cury, Moss, and Schor, 2013), suggesting that intestinal oxalate could be a marker for EH. In our mouse model, fecal and urine oxalate tended to correlate (please see new Figure 6D on page 39).

3. Lack of Urine oxalate data in these patients is a major weakness, and should be stressed more in the discussion1

We thank the reviewer for this comment. As above, we now provide mouse data on oxalate levels (new Figure 6 on page 39), which addresses the weakness pointed out by the reviewer.

Also, this limitation is stated in the Discussion as: “we did not have access to urinary oxalate or ascertain kidney stone history in our human IBD cohort, both critical variables to extend our conclusions”. However, previous studies have shown that intestinal oxalate is predominantly absorbed paracellularly *(Binder, 1974; Knauf et al., 2011; Saunders, Sillery, and McDonald, 1975)* along its concentration gradient; as such, colonic oxalate levels should correlate with both renal oxalate and with risk for CaOx stones.4. It is interesting that the oxalate degrading taxa and genes are reduced in patients at risk for EH, despite the metabolomics data that fecal oxalate is increased. This merits some discussion

We thank the reviewer for this comment. In patients at risk for EH, the abundance of ODP genes is increased (Figure S10), but the extent of gene expression is decreased (Figure 4). This contrast between gene abundance and expression is due to differential ODP transcription in particular microbes. Specifically, the increase of ODP gene abundances was due to the overrepresentation of *E. coli* that rarely transcribe ODP in vivo. In contrast, the decrease of ODP-gene expression is caused by loss of *O. formigenes*, which is the dominant microbe that transcribes this pathway. This point is now clarified in the Discussion.

5. There is no good evidence that oral Oxalobacter could be used to recolonize EH patients, or would effectively reduce urinary oxalate. This part of the discussion should be more guarded

We thank the reviewer for this comment. We agree that *O. formigenes* colonization in EH patients needs to be established in future studies. However, in EH, the intestinal lumen is an oxalate-rich environment, which supports *O. formigenes* colonization and oxalate degradation prior to absorption into the host circulation. Furthermore, a recent study (Canales and Hatch, 2017) of a surgery-induced EH rat model via Roux-en-Y gastric bypass (RYGB), indicated the potential of *O. formigenes* for treating EH. in the post-RYGB rat, *O. formigenes* strain OXWR achieved 100% colonization and decreased urinary oxalate by 74% compared to 39% in the sham-operated group.

As the reviewer suggested, we now have added both points into the Discussion.

Minor comment1. Be consistent in use of calcium-oxalate vs calcium oxalate. I would favor not using the hyphen.

Thank you. We have deleted the hyphen in “calcium-oxalate” throughout the manuscript, and changes are tracked in the document.

Reviewer 3In the manuscript entitled “Microbial contributions to oxalate metabolism in health and disease”, Liu et al. have performed a meta-analysis of human fecal multi’omics datasets focusing on describing the gene abundance and expression of oxalate degradation pathway (OPD) genes across these data sets, including IBD cohorts.The authors introduce the background and significance of this study highlighting that oxalate (systemic) toxicity is of significant importance as a risk factor of kidney stones and CKD. The authors then introduce that humans lack ODP but that gut microbes do, but that there are “gaps in our understanding of the role of the microbiota in diseases induced by oxalate toxicity”. The goals of the manuscript were to characterize ODP in human gut metagenomes and metatranscriptomes using pre-existing data sets.In Ffigure 1, the authors search gut metagenomes for genes in type 1 or type 2 ODPs and assign to microbial taxa. The main conclusion is that type I genes occur in both bacteria and fungi, type I only in bacteria. In Figure 2, the authors then show that most human gut samples have ODP genes and that type II genes are much more prevalent that type I. Focusing on type 2 ODP, in Figure 3 the authors then assign type 2 ODP genes to various bacterial taxa, mostly belonging to *E. coli* (Ec) and Oxalobacter formigenes (Of). Comparing metagenomics (MGX) and metatranscriptomics (MTX), the authors show that while for Of, gene content and transcript correlated, for Ec transcript as not often detected and that transcript were dominated in MTX byOf. The authors then in Figure 4 analyze data from IBD case and control cohorts (iHMP-IBD study), show that oxalate concentrations are higher in feces from people with IBD than control. Finally, in IBD compared to controls, the authors find decreased expression of type II ODP in IBD compared to controls. The reduced expression of ODP genes in IBD was assigned to Of (Figure 5).General CommentaryThe goals of this study characterizing ODP in human gut metagenomes and metatranscriptomes were met. This referee cannot formally critique the multi-omics methods or approach as not my area of expertise. I will critique this at the level of the novelty of the conclusions from this study, rather than a technical critique. There have been a number of microbiome compositional analysis associating oxalate nephrocalcinosis known of which were cited and this lack of citation misrepresents the field. The assignment of ODP genes to various taxa is a first comprehensive bioinformatics analysis, but not entirely novel, i.e. that organisms known to have this pathway were identified. The finding that ODP transcripts are dominated by Of compared to Ec is again “new” information but not necessarily unexpected as Of has an obligate requirement for oxalate and thus must express ODP genes for survival while Ec does not.

We appreciate the reviewer’s comments. Although we agree that some oxalatedegrading microbes have been individually characterized in vitro before, we believe our study provided novel insights with more depth and breadth, based on the following ideas:

1. We analyzed six ODPs that belong to two distinct types, and reached the novel conclusion that type I ODPs are encoded by both fungi and bacteria, but type II is exclusively encoded by bacteria, and that the human microbiome is associated with type II ODP.

2. Our study focuses on transcription of multiple oxalate-degrading enzymes in vivo using metatranscriptomics data. Such transcriptional evidence directly informs the functional potential of microbiota oxalate degradation. In contrast, the previous “microbiome compositional analysis” mentioned by the reviewer focused on microbiome population differences based on genetic-level changes (16S rRNA, and metagenomic sequencing). The lack of correlation between ODP genes and transcripts demonstrates that transcriptional data are critical when assessing the activity of these genes. Collectively,

this evidence illustrates the robustness of our analyses compared to prior studies that used gene-based measurements.

3. Although multiple oxalate-degrading microbes have been identified, as the reviewer pointed out, the relative importance of each taxon has been in debate for decades(Miller and Dearing, 2013). Therefore, there have been both academic and commercial efforts using several other bacteria to alleviate oxalate toxicity (Cho, Gebhart, Furrow, and Lulich, 2015; Ellis, Shaw, Jackson, Daniel, and Knight, 2015; Lieske, 2017; Miller, Choy, Penniston, and Lange, 2019; Ticinesi, Nouvenne, and Meschi, 2019). We now provide clear evidence that demonstrates the dominant role of *O. formigenes* in oxalate degradation within the human microbiota. Moreover, we demonstrate potential clinical indications (e.g. Crohn’s disease of the ileocolonic phenotype, and Ulcerative colitis) when *O. formigenes* is lost.

Entirely Of the most novel findings of the study are the inverse correlation between Of ODP gene expression and oxalate levels in IBD. This raises interesting questions whether dysfunction in Of ODP gene expression may lead to increased gut oxalate in IBD but not healthy guts. Furthermore, why would Of not express these genes in these situations?

We thank the reviewer for this comment. Indeed, our findings indicate for the first time that microbiota-based oxalate degradation is significantly associated with the elevated fecal oxalate levels in IBD patients. The ODP of *O. formigenes*, which was the dominating ODPtranscribing microbe in health, was detected at significantly lower levels in IBD patients. The biological nature of *O. formigenes* as an oxalate autotroph (Cornick and Allison, 1996), requires that it continually transcribe ODP for both carbon source and for energy, leading to ATP production (Cornick and Allison, 1996). Therefore, the lack of *O. formigenes* ODP transcription most likely indicates its absence in IBD patients. Previous studies have demonstrated its loss of colonization is associated with antibiotic use, which is common in IBD patients (Liu et al., 2017). Our study now raises testable hypotheses that common treatments including antibiotics and other medications with antibacterial activities have caused the loss of these bacteria in IBD patients.

In general, I am under impressed by this manuscript. The description of ODP pathways in metagenomes and the expression by Oxalobacter formigens and *E. coli* is not particularly unexpected. The inverse relationship between oxalate levels in IBD and Oxalobacter expression is interesting but again I can't equate this finding with the level of findings of this journal.

We appreciate the reviewer’s time and comments. We have provided evidence (above) of how our study has provided novel insights in this area in both health and disease, and have included new data from a mouse model supporting the important in vivo role of *O. formigenes* in oxalate homeostasis (new Figure 6 in on page 39).

References:

Binder, H. J. (1974). Intestinal oxalate absorption. *Gastroenterology, 67*(3), 441-446.

Canales, B. K., and Hatch, M. (2017). Oxalobacter formigenes colonization normalizes oxalate excretion in a gastric bypass model of hyperoxaluria. *Surg Obes Relat Dis, 13*(7), 1152-1157. doi:10.1016/j.soard.2017.03.014

Cho, J. G., Gebhart, C. J., Furrow, E., and Lulich, J. P. (2015). Assessment of in vitro oxalate degradation by *Lactobacillus* species cultured from veterinary probiotics. *Am J Vet Res, 76*(9), 801-806. doi:10.2460/ajvr.76.9.801

Cornick, N. A., and Allison, M. J. (1996). Assimilation of oxalate, acetate, and CO2 by Oxalobacter formigenes. *Can J Microbiol, 42*(11), 1081-1086. doi:10.1139/m96-138

Cury, D. B., Moss, A. C., and Schor, N. (2013). Nephrolithiasis in patients with inflammatory bowel disease in the community. *Int J Nephrol Renovasc Dis, 6*, 139-142. doi:10.2147/IJNRD.S45466

Ellis, M. L., Shaw, K. J., Jackson, S. B., Daniel, S. L., and Knight, J. (2015). Analysis of Commercial Kidney Stone Probiotic Supplements. *Urology, 85*(3), 517-521. doi:https://doi.org/10.1016/j.urology.2014.11.013

Knauf, F., Ko, N., Jiang, Z., Robertson, W. G., Van Itallie, C. M., Anderson, J. M., and Aronson, P. S. (2011). Net intestinal transport of oxalate reflects passive absorption and SLC26A6-mediated secretion. *J Am Soc Nephrol, 22*(12), 2247-2255. doi:10.1681/ASN.2011040433

Lieske, J. C. (2017). Probiotics for prevention of urinary stones. *Annals of translational medicine, 5*(2), 29-29. doi:10.21037/atm.2016.11.86

Liu, M., Koh, H., Kurtz, Z. D., Battaglia, T., PeBenito, A., Li, H.,... Blaser, M. J. (2017). Oxalobacter formigenes-associated host features and microbial community structures examined using the American Gut Project. *Microbiome, 5*(1), 108. doi:10.1186/s40168-017-0316-0

Miller, A. W., Choy, D., Penniston, K. L., and Lange, D. (2019). Inhibition of urinary stone disease by a multi-species bacterial network ensures healthy oxalate homeostasis. *Kidney Int, 96*(1), 180-188. doi:10.1016/j.kint.2019.02.012

Miller, A. W., and Dearing, D. (2013). The metabolic and ecological interactions of oxalate-degrading bacteria in the Mammalian gut. *Pathogens, 2*(4), 636-652. doi:10.3390/pathogens2040636

Saunders, D. R., Sillery, J., and McDonald, G. B. (1975). Regional differences in oxalate absorption by rat intestine: evidence for excessive absorption by the colon in steatorrhoea. *Gut, 16*(7), 543-548. doi:10.1136/gut.16.7.543

Ticinesi, A., Nouvenne, A., and Meschi, T. (2019). Gut microbiome and kidney stone disease: not just an Oxalobacter story. *Kidney Int, 96*(1), 25-27. doi:https://doi.org/10.1016/j.kint.2019.03.020